# Spectral signatures of excess-proton waiting and transfer-path dynamics in aqueous hydrochloric acid solutions

Florian N. Brünig ⬤ [1], Manuel Rammler ⬤ [1], Ellen M. Adams[2], Martina Havenith[2] & Roland R. Netz ⬤ [1✉]

The theoretical basis for linking spectral signatures of hydrated excess protons with microscopic proton-transfer mechanisms has so far relied on normal-mode analysis. We introduce trajectory-decomposition techniques to analyze the excess-proton dynamics in ab initio molecular-dynamics simulations of aqueous hydrochloric-acid solutions beyond the normal-mode scenario. We show that the actual proton transfer between two water molecules involves for relatively large water-water separations crossing of a free-energy barrier and thus is not a normal mode, rather it is characterized by two non-vibrational time scales: Firstly, the broadly distributed waiting time for transfer to occur with a mean value of 200–300 fs, which leads to a broad and weak shoulder in the absorption spectrum around 100 cm$^{-1}$, consistent with our experimental THz spectra. Secondly, the mean duration of a transfer event of about 14 fs, which produces a rather well-defined spectral contribution around 1200 cm$^{-1}$ and agrees in location and width with previous experimental mid-infrared spectra.

[1] Freie Universität Berlin, Department of Physics, 14195 Berlin, Germany. [2] Ruhr-Universität Bochum, Department of Physical Chemistry II, 44780 Bochum, Germany. ✉email: rnetz@physik.fu-berlin.de

The motion of excess protons in aqueous solution is fundamental for many biological and chemical processes. The excess-proton diffusivity is significantly higher compared to other monovalent cations in water[1,2], since the excess proton exchanges its identity with water hydrogens during the diffusion process[3,4]. Grotthus hypothesized a similar process over two centuries ago[5,6], but a detailed understanding of the proton-transfer dynamics between water or other molecules remains difficult to date due to the multitude of time scales involved and the only indirect experimental evidence.

Infrared (IR) spectroscopy in the THz and mid-IR regimes is a powerful tool to explore the ultrafast dynamics of water and aqueous ion solutions. For example, the prominent absorption peak around $200 \, \text{cm}^{-1}$ of bulk water is dominated by first-solvation-shell dynamics, whereas motion involving the second solvation shell contributes most significantly below $80 \, \text{cm}^{-1}$ (2.4 THz)[7,8]. Furthermore, so-called 'rattling' modes for strongly hydrated ions lead to characteristic absorption features, while for weakly hydrated ions vibrationally induced charge fluctuations are dominant[9,10], as suggested by dissecting simulation spectra into contributions from different solvation shells[11,12].

IR spectroscopy has proven particularly useful for the study of the ultrafast dynamics of the excess proton in aqueous solution[13,14]. Due to their low pH value, aqueous hydrochloric acid (HCl) solutions are perfect model systems to study excess-proton dynamics, since HCl dissociates readily in water and gives rise to a large number of highly mobile protons that are only weakly coordinated with neutralizing chloride ions and therefore behave as if added in excess. The characteristic continuum band in the IR absorption spectrum, located between the water-bending mode around $1650 \, \text{cm}^{-1}$ and the water-stretching mode around $3300 \, \text{cm}^{-1}$, has long been known and led to the hypothesis of the Zundel state, i.e., two water molecules symmetrically sharing the excess proton[15]. This model has been challenged by a contrasting picture, the Eigen state, which is a hydronium ion caged symmetrically by three water molecules[16]. Ever since these idealized structures have been proposed, their relative stability has been controversially debated[13,17–27]. It is now known that neither the idealized Zundel nor the Eigen states are realistic structural representations and that the excess proton mostly resides slightly asymmetrically shared between two water molecules, in the 'special pair' state, which geometrically can be interpreted as a 'distorted Zundel' state or a 'distorted Eigen' state[20,23,24,26–28].

While simulations can reproduce most experimental spectroscopic signatures, the understanding of the proton transfer mechanism requires model building based on and guided by simulations. It is generally accepted that proton transfer involves consecutive transitions between states that can be viewed as more Eigen-like and more Zundel-like and have fast interconversion times[24,26,29]. That the excess proton diffusion involves the crossing of free-energetic barriers follows from the experimentally known Arrhenius behavior of the excess-proton conductivity[28,30–32]. Similar to the above-mentioned discussion on the relative stability of the Eigen and Zundel states, it remains debated whether the Zundel state is the transition state between two Eigen states or the opposite is the case, i.e., whether the Eigen state is the transition state between two Zundel states[23,24,27]. Theoretical models for the spectroscopic signatures of the hydrated proton motion so far relied on normal-mode calculations and have explained many aspects of experimental linear absorption[33–35] as well as 2D IR spectra[25,26]. However, normal modes by construction cannot deal with the thermally activated transfer of an excess proton over a free-energy barrier, since this corresponds to an unstable mode with a negative free-energy curvature along the transfer reaction coordinate[36]. It is clear that

such proton-barrier-transfer events will make a sizable spectroscopic contribution, since they involve fast motion of a highly charged object over relatively large distances. From this follows that an excess-proton transition state, which corresponds to a free energy maximum and thus occurs with a small probability, nevertheless can make a dominant contribution to the spectrum, which would lead to characteristic differences between experimental spectra and normal-mode theory predictions. Indeed, it has been noted that the normal-mode spectra computed from instantaneous configurations do not explain all experimental spectral signatures associated with the excess proton in water[19,25–27], in particular of the proton-transfer dynamics[37]. It was recently shown that proton transfer in the $H_5O_2^+$ cationic complex gives rise to two distinct spectroscopically relevant time scales that cannot be captured by normal-mode analysis[38].

In general, the transfer of a particle with mass $m$ over an energy barrier with negative curvature $k < 0$ corresponds to an unstable mode. The dynamics of such a barrier-crossing is not characterized by a vibrational time scale, which according to a harmonic oscillator model could erroneously be written as $\tau \simeq \sqrt{m/|k|} \simeq \sqrt{mL^2/U_0}$, where $U_0$ is the barrier height and $L$ is the barrier width, but rather by two other time scales, namely the mean transfer-waiting time $\tau_{TW}$ and the mean transfer-path time $\tau_{TP}$. $\tau_{TW}$ is the average waiting time before a transfer event occurs and $\tau_{TP}$ is the average time of the actual transfer path over the energy barrier. The former one scales exponentially with the barrier height $U_0$, $\tau_{TW} \sim \exp(U_0/k_B T)$[36,39], whereas the latter one scales logarithmically with the barrier height $U_0$, $\tau_{TP} \sim \log(U_0/k_B T)$[40–43]. In essence, it is not clear with current theoretical methodology what the spectroscopic signature of excess-proton transfer events in aqueous solutions is and whether the continuum band stems just from vibrations in metastable states or whether transfer reactions over barriers are involved. Thus, a theoretical approach that is complementary to normal modes and can handle proton-transfer events that involve free-energy barriers is needed.

In this study, we investigate the excess-proton dynamics in aqueous HCl solutions at ambient conditions using ab initio molecular dynamics (MD) simulations at the Born–Oppenheimer level and experimental THz/Fourier-transform infrared (THz/FTIR) measurements. Our simulated IR difference absorption spectra compare well to our experimental data in the THz regime as well as to literature data in the mid-IR regime. By projecting the excess-proton dynamics onto the two-dimensional coordinate system spanned by the proton position along the axis connecting the two closest water oxygens $d$ and the oxygen distance $R_{OO}$[44,45], the excess-proton trajectories and their spectral signatures are subdivided into three contributions with distinct time scales, as illustrated in Fig. 1a. The fastest time scale, $\tau_{NM}$, reflects vibrations when the excess proton transiently forms a solvated $H_3O^+$ molecule which is asymmetrically solvated in a special pair. It is well captured by a normal-mode description and has been amply discussed in literature[19,25,26,37,46]. The other two spectral signatures, stemming from proton transfer events, are the focus of this study. The associated time scales $\tau_{TW}$ and $\tau_{TP}$ are directly obtained from our simulated excess proton trajectories using a multidimensional path analysis. While the transfer-waiting time $\tau_{TW}$ of aqueous proton-transfer events has been studied recently[47], the identification of both $\tau_{TW}$ and $\tau_{TP}$ in simulated and experimental spectra is a main result of this work. We find a mean transfer-waiting time of $\tau_{TW} = 200–300$ fs depending on HCl concentration, which in our experimental THz spectra shows up as a broad weak shoulder around $100 \, \text{cm}^{-1}$, that is partially overlaid by the absorption due to rattling chloride anions at about $150 \, \text{cm}^{-1}$[9,11]. The mean transfer-path time, from simulations obtained as $\tau_{TP} = 14$ fs, produces a spectroscopic signature

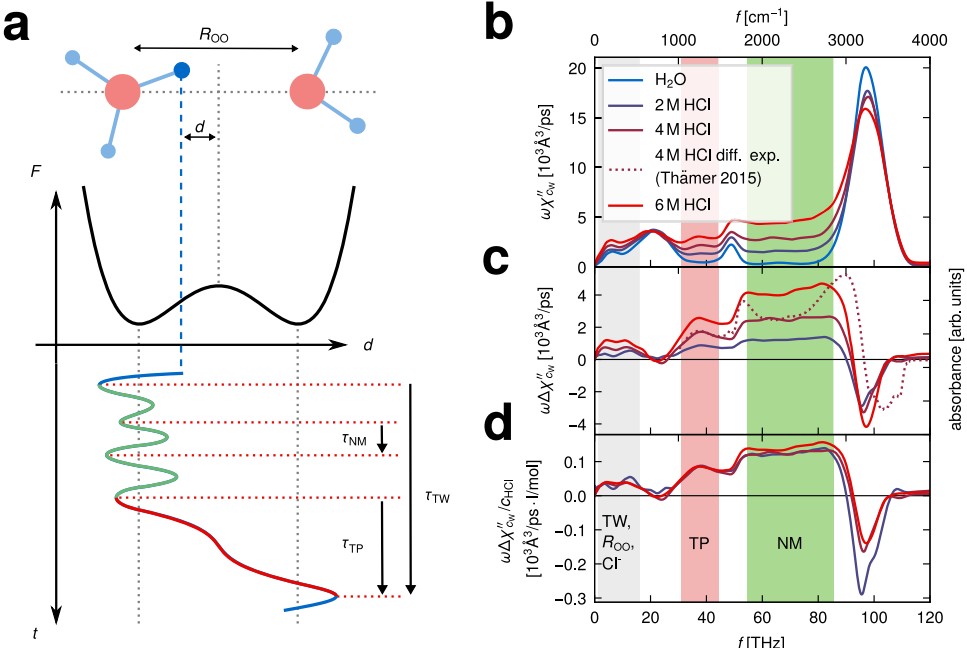

**Fig. 1 Time scales of excess-proton dynamics and simulated absorption spectra. a** Schematic trajectory of an excess proton that transfers between two water molecules, together with a schematic free energy profile $F(d)$ that exhibits a barrier and is representative of a relatively large oxygen-oxygen separation $R_{OO}$. Three time scales characterize the proton trajectory, the normal-mode vibrational period of the solvated transient $H_3O^+$, $\tau_{NM}$, the transfer-path time, $\tau_{TP}$ and the transfer-waiting time, $\tau_{TW}$, where $\tau_{TW} > \tau_{TP} > \tau_{NM}$. An animation is shown online https://fu-berlin.eu.vbrickrev.com/sharevideo/df2d94a4-6e7f-499a-a256-17d73b6124e4. **b** Infrared (IR) absorption spectra obtained from ab initio molecular dynamics (MD) simulations of pure water (blue solid line) and hydrochloric acid (HCl) solutions at various concentrations (dark purple: 2 M, purple: 4 and red: 6 M). The spectra are divided by the water molecular number concentration $c_W$. **c** Difference spectra between the three HCl spectra and the water spectrum, obtained from the spectra in b. The purple dotted line shows an experimental difference spectrum of HCl at 4 M[14], rescaled in height to match the simulation results. **d** The simulated difference spectra (as shown in **c**) divided by the HCl concentrations $c_{HCl}$. Three distinct spectral regions are shaded in different colors, that are identified with different excess-proton dynamic processes: transfer waiting (TW, gray), transfer paths (TP, red), and normal modes (NM, green). The transfer-waiting time is close to the chloride-ion (Cl⁻) rattling time and the oxygen vibrational time in local $H_5O_2^+$ complexes that is described by the $R_{OO}$ coordinate.

around 1200 cm⁻¹, which is well captured in experimental mid-IR spectra[14,20,23,24,26]. Note that in our simulations the proton transfer becomes barrier-less for small water separation and thus includes the highly anharmonic normal-mode vibration of Zundel-like configurations, which have already been analyzed theoretically[19,25,37,46]. In the THz regime our experimental difference spectra show an additional prominent peak around 300–400 cm⁻¹, in good agreement with our simulation data, which is demonstrated to be caused by the coupling of the excess-proton motion to the relative oscillations of the two flanking water molecules in transient $H_5O_2^+$ complexes. Proton transfer events between water molecules are frequently followed by a few immediate back-and-forth transfer events, which is a consequence of non-Markovian effects[48] that have to do with the slowly changing solvation structure around the excess proton. Although these transfer events are therefore not always productive in the sense that they lead to large-scale diffusion of the excess proton, they nevertheless give rise to pronounced experimental spectroscopic signatures and therefore need to be included in the analysis.

## RESULTS

**Infrared and THz spectra of HCl solutions.** Within linear spectroscopy, the energy absorption rate of incident light with frequency $f = \omega/(2\pi)$ is proportional to the imaginary part of the dielectric susceptibility and given by $\omega\chi''(\omega)$. IR power spectra are obtained from ab initio MD simulations of water (blue solid line) and HCl solutions at three concentrations between 2 and 6 M

(purple to red solid lines) and are shown in Fig. 1b. The spectra are divided by the water molecular number concentration $\omega\chi''_{c_W} = \omega\chi''/c_W$. Simulation details are provided in the "Methods" section. All IR spectra show the characteristic features of pure water spectra, which are the prominent OH-stretching peak around 3300 cm⁻¹, the HOH-bending mode around 1650 cm⁻¹ and librational modes in the far IR regimes between 200 and 800 cm⁻¹. The IR spectra of HCl solutions additionally show a broad continuum between the bending and the stretching peaks, from 2000 to 3000 cm⁻¹, and a broad peak at around 1200 cm⁻¹, both of which are commonly interpreted as to reflect the excess-proton dynamics[14,23,49]. Furthermore, additional features are observed below 800 cm⁻¹, that are shown in Fig. 2 in comparison to our experimental THz spectra and will be discussed further below.

The simulated difference spectra in Fig. 1c (solid lines) clearly demonstrate three distinct regions (color shaded), that relate to distinct time scales of the excess-proton dynamics and will in this work be identified as transfer-waiting (TW, gray), transfer-path (TP, red) and normal-mode contributions (NM, green). We obtain rather good agreement with the experimental difference spectrum for 4 M HCl[14], which was scaled to match the height of the simulated IR 1200 cm⁻¹ peak, see Supplementary Fig. 5 for a comparison of different experimental data. Our simulated 4 M difference spectrum in Fig. 1c does not reproduce the local maximum of the experimental difference spectra around 1750 cm⁻¹, which is interpreted as the acid-bend band, i.e., a blue shift of the bending mode in $H_3O^+$ compared to water, and also not the shape of the experimental acid-

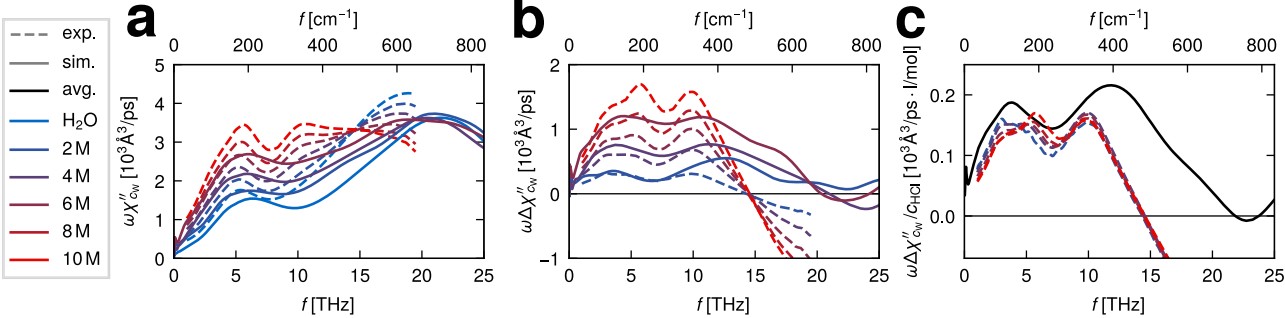

**Fig. 2 Experimental absorption spectra. a** Experimental THz/Fourier-transform infrared (THz/FTIR) absorption spectra of HCl solutions at various concentrations (colored broken lines for 2 M to 10 M), compared to literature data of pure water[74] (blue broken line). The experimentally measured extinction spectra have been converted into energy absorption spectra using the Kramers–Kronig relation, no amplitude adjustment is used in the comparison with the ab initio spectra (solid lines). **b** Experimental and ab initio molecular dynamics (MD) difference spectra derived from the results given in a and plotted in the respective colors and line styles. **c** The experimental difference spectra (shown in b) are divided by the HCl concentration $c_{HCl}$ (colored broken lines) and compared to the average of the simulated difference spectra after dividing by $c_{HCl}$ (black solid line).

stretch signature around 3000 cm$^{-1}$ [14,19,20,23,50]. The reason for this disagreement is unclear, we note that the normalization of spectra when calculating difference spectra is a subtle issue, see Supplementary Note 3 for a discussion.

Figure 1d shows that the three simulated HCl difference spectra divided by the HCl concentrations $c_{HCl}$ are nearly indistinguishable. This clearly indicates that the spectroscopic features are due to single-proton dynamics and that collective proton effects as well as proton-chloride coupling effects, which would scale non-linearly in the HCl concentration, are minor. This is an important finding and justifies our theoretical analysis of single excess-proton motion in this work.

In order to investigate the intermolecular vibrational dynamics of water, solvated protons, and chloride ions, we experimentally measure THz absorption spectra for HCl concentrations of 2 M, 4 M, 6 M, 8 M, and 10 M. For quantitative comparison with our simulation data, the experimentally measured extinction spectra are converted into energy absorption spectra using the Kramers–Kronig relation, details are described in the "Methods" section and in Supplementary Methods 1. The experimental THz/FTIR spectra are shown in Fig. 2a in the range 0–650 cm$^{-1}$ (colored broken lines) together with a literature spectrum of pure water (blue broken line) and are compared to the available simulated spectra (solid lines). Again, all experimental and simulated spectra are divided by the respective water concentration $c_w$. One notes the good agreement between the experimental and simulation spectra below 400 cm$^{-1}$, which is noteworthy since the spectral amplitudes are not rescaled or adjusted. However, the reason of the disagreement for larger wave numbers is not clear. All spectra show a prominent peak at 200 cm$^{-1}$. Difference spectra of the experimental data with respect to the pure water spectrum are shown in Fig. 2b (broken lines) and again compared to the available simulated difference spectra (solid lines). Two peaks dominate the difference spectra, one around 150–200 cm$^{-1}$ and one around 300–400 cm$^{-1}$. The experimental difference spectra scale linearly with HCl concentration, which is demonstrated in Fig. 2c, where the difference spectra are divided by the HCl concentrations $c_{HCl}$. For comparison, the simulated difference spectra divided by $c_{HCl}$, already presented in Fig. 1d, are averaged over the three HCl concentrations and shown as a black solid line. The linear scaling of the experimental spectra with HCl concentration reconfirms that the difference spectra are related to single-ion behavior and that collective ion effects are negligible, in agreement with previous observations[13,14]. In essence, two different processes at 150–200 cm$^{-1}$ and 300–400 cm$^{-1}$ are clearly indicated by our experimental and simulated spectra and

will be interpreted by our spectral trajectory-decomposition techniques. In the remainder we analyze exclusively the 6 M solution, which provides the best proton statistics.

**Excess-proton trajectories and spectra**. Excess protons constantly change their identity as they move through the HCl solution. Each identity change introduces a spurious discontinuity in the excess-proton trajectory, which does not actually correspond to charge transport and therefore is spectroscopically irrelevant. In order to extract continuous excess-proton trajectories from our simulations, we use a dynamic criterion as illustrated in Fig. 3a (that our extracted excess-proton trajectories are spectroscopically meaningful we will a posteriori demonstrate by comparison of spectra calculated from excess proton trajectories with spectra calculated from the complete simulation system). Each proton is assigned to its closest oxygen atom at each time step. Whenever three protons are assigned to the same oxygen, thereby forming a hydronium ion, all of them are registered as excess-proton candidates. That means, for the generation of continuous excess-proton trajectories, we do not select the hydrogen with the largest separation from the oxygen, which would lead to fast switching of the excess proton identity, the so-called 'special pair dance' of hydronium with its surrounding water molecules[27,51]. Rather, if during the simulation an excess-proton candidate becomes assigned to a different oxygen and thus transfers to a neighboring water, it is selected as an excess proton for the entire time during which it was part of any hydronium ion[27]. Note that the spectral effects of the rattling of the excess-proton candidates within one hydronium ion, i.e., the 'special pair dance', are in some of our calculations below included by taking into account the flanking water molecules in the calculation of spectra, but do not show significant spectral signatures. Excess protons that are coordinated with a chloride anion as the second nearest neighbor are neglected from our analysis. This does not influence our excess-proton spectra, since even for the highest acid concentration of 6 M, only 5% of all configurations are of this type, as demonstrated in Supplementary Table 3. Note, however, that the fraction of protons coordinated with chloride ions increases significantly at higher concentrations[52]. Our procedure for calculating continuous excess-proton trajectories is discussed in further detail in Supplementary Methods 2.

The excess-proton trajectories are described by the two-dimensional coordinate system defined within local transient $H_5O_2^+$ complexes consisting of the excess proton and its two nearest water molecules, as illustrated in the right part of Fig. 3a. The coordinates are the instantaneous distance between the two

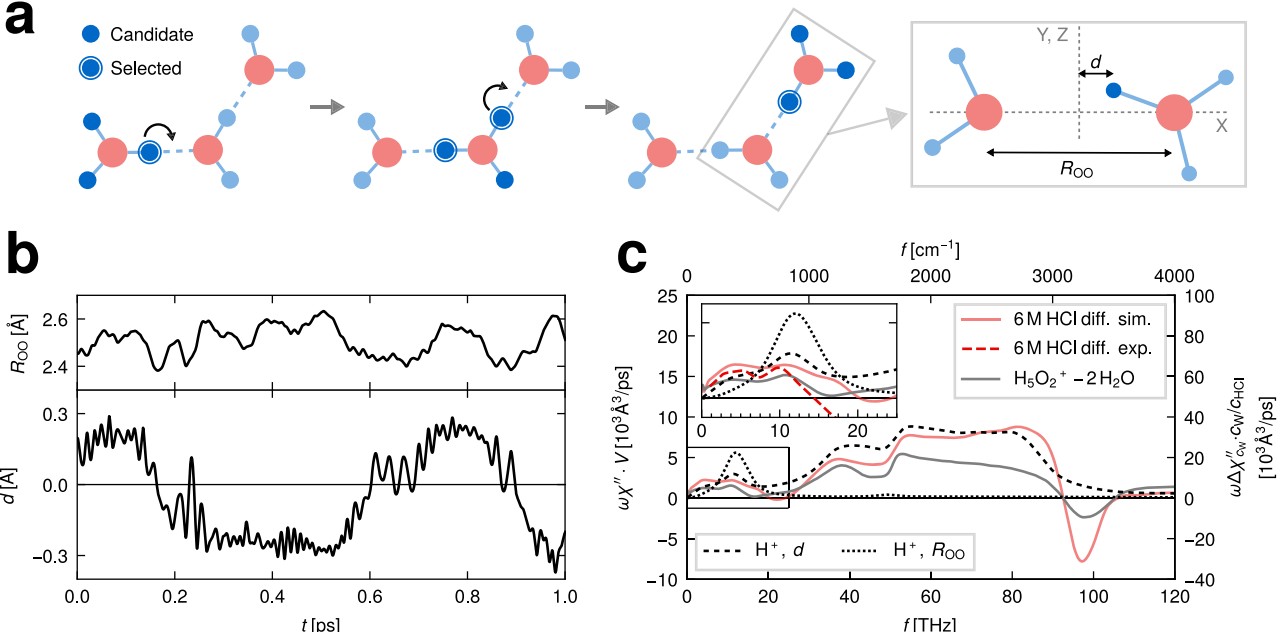

**Fig. 3 Excess-proton trajectories and spectra. a** Illustration of the method used to extract continuous excess-proton trajectories from ab initio molecular dynamics (MD) simulations. From the three protons in a hydronium ion, the one that will later transfer to a neighboring water is identified as excess proton. To the right the coordinates $d$ and $R_{OO}$ are defined. **b** Example trajectories of the $d$ and $R_{OO}$ coordinates. **c** Power spectra of the $d$ and $R_{OO}$ coordinates are shown as broken and dotted black lines, respectively (vertical axis on the left). Also shown are the simulated difference spectrum of the 6 M HCl solution, divided by the HCl concentration $c_{HCl}$ and multiplied by the water concentration $c_W$ (red solid line), and the simulated difference spectrum between a transient $H_5O_2^+$ complex in HCl solution and two hydrogen-bonded water molecules in pure water (gray solid line), using the vertical axis on the right. The inset shows a zoom into the THz regime, which additionally shows our 6 M experimental THz/FTIR difference spectrum (red broken line).

oxygen atoms $R_{OO}$ and the excess proton's distance from the midplane $d$[45]. The state for $d = 0$, where the excess proton is in the middle between the oxygens, will be later used to define the transition state of the proton transfer between the two flanking water molecules. Figure 3b shows an example excess-proton trajectory from our ab intio MD simulations in terms of the $R_{OO}$ and $d$ coordinates. While the motion along the two coordinates is strongly correlated, as we will show later, the $d$ trajectory shows fast oscillatory components that are much weaker in the $R_{OO}$ trajectory.

The power spectra of the excess-proton trajectories, averaged over all excess protons in the solution, are shown in Fig. 3c for the $d$ coordinate as a black broken line and for the $R_{OO}$ coordinate as a black dotted line, in the calculations we assume a bare charge of 1 e for the excess proton (left axis). We compare with the simulated difference spectrum of the 6 M HCl solution (red solid line), which is multiplied by the water concentration $c_W$ and divided by HCl concentration $c_{HCl}$ and thus is normalized per excess proton (right axis). The qualitative agreement between the two spectra (black broken and red solid lines) is very good up to an overall scaling factor of roughly four, which reflects polarization enhancement due to neighboring water molecules. The good agreement indicates that the difference spectrum of an HCl solution is proportional to the spectrum of the highly IR-active excess proton in terms of its coordinate $d$[47]. In other words, the HCl-solution difference spectrum reports on the excess-proton motion relative to the two flanking water oxygens and can therefore be used to investigate proton-transfer dynamics. In turn, the analysis of excess-proton trajectories allows to reveal the microscopic mechanism causing the signatures of HCl-solution difference spectra, which is a central validation of the trajectory-decomposition technique used in our study. In contrast, the dynamics of $R_{OO}$, i.e., the vibrations of the water molecules in the $H_5O_2^+$ complex, black dotted line in Fig. 3c, gives rise to a single

spectral feature around 400 cm$^{-1}$, which turns out to be present also in the simulated and experimental HCl-solution spectra, as will be discussed below.

To check for the effect of the two water molecules that flank the excess proton on the difference spectrum, we also calculate the IR spectrum of transient $H_5O_2^+$ complexes, as done previously[22,53] and presented in more detail in Supplementary Fig. 11. To construct a difference spectrum, we subtract from the $H_5O_2^+$ spectrum the spectrum of hydrogen-bonded water-molecule pairs obtained from the pure-water ab initio MD simulation. The resulting difference spectrum in Fig. 3c (gray solid line, right scale) is reduced by a factor of roughly two compared to the difference spectrum of the entire HCl solution (red solid line) but otherwise agrees in shape rather nicely. Compared to the spectrum of the isolated excess proton (broken line, left scale) we observe an amplification by a factor of roughly two, but no essential spectral shape change. We conclude that the flanking water molecules and in particular the 'special pair dance' with further solvating water molecules does not modify the spectrum of the excess proton in an essential way. The amplification of the complete HCl-solution difference spectrum compared to the $H_5O_2^+$ difference spectrum (red and gray solid lines, respectively) we rationalize by polarization enhancement effects of water molecules that solvate the $H_5O_2^+$ complex.

A few spectral contributions that are not included in the excess proton power spectrum (black broken line in Fig. 3c) deserve mentioning: (i) Dynamics orthogonal to the connecting axis of the oxygens are shown to be small in Supplementary Fig. 11. (ii) The chloride motion is shown below to contribute only slightly and at low frequencies to the spectrum. iii) The translation and rotation of the internal $H_5O_2^+$ coordinate system relative to the lab frame is in Supplementary Figs. 12 and 13 shown to only give a small spectral contribution. We thus conclude that the IR difference spectrum between HCl solutions and pure water

 **5**

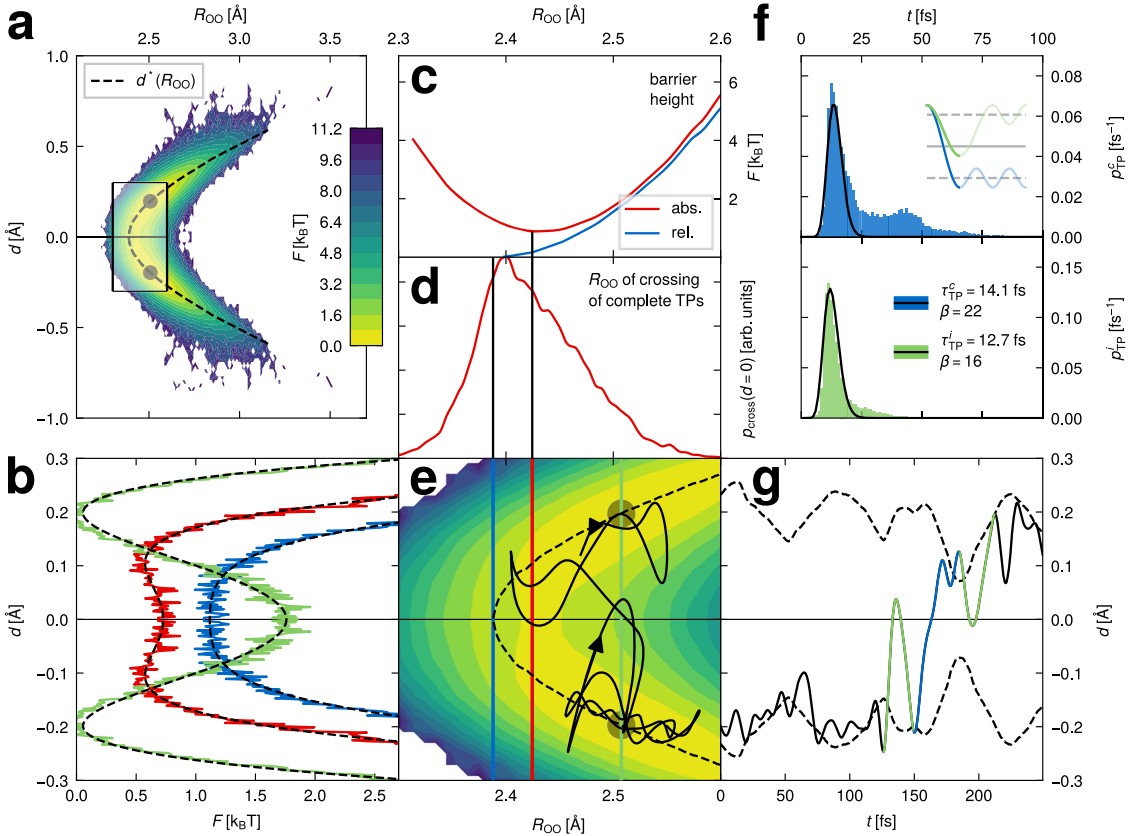

**Fig. 4 2D excess-proton trajectory analysis. a** The two-dimensional (2D) free energy of the excess protons in 6 M HCl solution for the $(d, R_{OO})$ coordinates defined in the inset of Fig. 3a. The shaded area is enlarged and shown in e. The gray dots denote the positions of the global minima of the 2D free energy. The minima for fixed $R_{OO}$, i.e., the most likely proton locations, $d^*(R_{OO})$, are indicated by a black broken line. **b** Cuts through the free energy in a for $R_{OO} = 2.39$ Å, where the barrier just vanishes (blue solid line), $R_{OO} = 2.42$ Å, where the absolute barrier height is minimal (red solid line) and $R_{OO} = 2.51$ Å, for which the global minima of the 2D free energy are obtained (green solid line). **c** The absolute free energy at $d = 0$ (red solid line) and the barrier height relative to the minima at fixed $R_{OO}$ located at $d^*(R_{OO})$ (blue solid line). **d** Distribution of $R_{OO}$ positions at which complete transfer paths cross the $d = 0$ midplane. **e** Zoom into the free energy shown in a. The vertical colored lines indicate the cuts through the free energy shown in b. An example trajectory from the ab initio molecular dynamics (MD) simulation is shown as a black solid line. **f** Path-duration distributions of complete (blue, times defined from $d^*$ to $-d^*$) and incomplete transfer paths (green, times defined from $d^*$ to $d = 0$). Fits according to Eq. (1) are shown as black solid lines. **g** Time course of the example trajectory along $d$, same as shown in e, with complete and incomplete transfer paths indicated as blue and green lines, respectively. The two branches of the $R_{OO}$-dependent minimal energy position $d^*[R_{OO}(t)]$ are shown as black broken lines.

reports very faithfully on the excess-proton dynamics, apart from an overall amplification factor.

**2D excess-proton trajectory analysis.** Figure 4a shows the two-dimensional (2D) free energy for 6 M HCl obtained from the negative logarithm of the distribution function of the continuous excess-proton trajectories as a function of the coordinates $d$ and $R_{OO}$, a blow up of the shaded area is given in Fig. 4e. By definition, the free energy is symmetric with respect to the midplane at $d = 0$, which separates two global minima at $R_{OO} = 2.51$ Å and $d = \pm 0.2$ Å. These minima, highlighted as gray dots in Fig. 4a, correspond to states where the excess proton is asymmetrically shared between the two flanking water molecules. The transition between these minima, i.e., the proton transfer, is, therefore, a barrier-crossing process in the two-dimensional plane spanned by $R_{OO}$ and $d$.

Figure 4b shows cuts through the free energy for constant $R_{OO}$ along $d$, each fitted to a quartic expression $F(d) = F_{d=0}(1 + \gamma_2 d^2 + \gamma_4 d^4)$ shown as black broken lines. Details are reported in Supplementary Methods 3. For negative $\gamma_2$, two minima at $d^* = \pm\sqrt{-\gamma_2/(2\gamma_4)}$ are separated by a barrier at $d = 0$, which correspond to the optimal proton asymmetry for a given value

of $R_{OO}$ and are determined by the parabolic function $d^*(R_{OO})$, which is plotted as a black broken line in Fig. 4a, e. The cuts in Fig. 4b are shown for $R_{OO} = 2.39$ Å, where the barrier just vanishes (blue solid line), $R_{OO} = 2.42$ Å, where the absolute barrier height is minimal and which contains the transition state in the $d$-$R_{OO}$ plane at $d = 0$ (red solid line), and $R_{OO} = 2.51$ Å, which contains the global minima of the 2D free energy (green solid line).

The absolute free energy at $d = 0$ is plotted in Fig. 4c as a red solid line and compared to the barrier height relative to the $R_{OO}$-dependent minima at $d^*(R_{OO})$ (blue solid line). The minimal absolute barrier free energy of $0.9\, k_BT$ (red line), located at $R_{OO} = 2.42$ Å, defines the transition state; for $R_{OO} = 2.51$ Å, for which the most probable excess-proton state is obtained, the barrier has a moderate absolute height of $1.8\, k_BT$, suggesting that proton transfer is not excluded for this value of $R_{OO}$. Note that for $R_{OO} < 2.39$ Å the relative barrier height vanishes and thus a symmetrically shared excess proton is most likely.

Next, to decompose the excess proton trajectories into segments where the excess proton moves around the local free energy minima and where a transfer across the midplane happens, transfer-path start and end points need to be defined. For this we use the most likely proton location $d^*(R_{OO})$ (black

broken lines in Fig. 4a, e, g). The start of a transfer path is thus defined as the last crossing of $d^*(R_{OO})$ on one side of the midplane at $d = 0$ and the end of a transfer path as the first crossing of $d^*(R_{OO})$ on the other side of the midplane at $d = 0$. The transfer paths are slightly extended forward and backward in time to the points where the velocity along $d$ vanishes, the so-called turning points, in order to be consistent with the analytical theory presented in[38]. An example trajectory is shown in Fig. 4e (thin black line) in the $(d, R_{OO})$ plane, the corresponding time-dependent position $d$ is given in Fig. 4g, where the transfer path is highlighted in blue. Many transfer attempts are unsuccessful and lead to incomplete transfer paths, where the excess proton crosses the mid-plane $d = 0$ but does not reach to the minimal free energy state $d^*(R_{OO})$ on the other side. Two incomplete transfer paths are shown in green in Fig. 4g and for consistency are also extended to their turning points. Transfer-path-time distributions are given in Fig. 4f; with our definitions used, incomplete transfer paths turn out to be slightly faster. In total, there are about as many incomplete ($n = 14877$) as complete transfer paths ($n = 14357$), meaning that about half of all excess protons reaching the midplane $d = 0$ actually transfer from one water molecule to the other. The main peaks in the distributions are fitted by the Erlang distribution[54–56]

$$p_{TP}(t) = \frac{t^{\beta-1}}{(\beta-1)!}\left(\frac{\beta}{\tau_{TP}}\right)^\beta e^{-\beta t/\tau_{TP}}, \tag{1}$$

with the mean transfer-path time defined by $\tau_{TP}$, shown as black solid lines in Fig. 4f, the fit parameters are given in the legend.

The distribution of transition states in Fig. 4d, i.e., the $R_{OO}$ position at which complete transfer paths cross the midplane at $d = 0$, is rather broad and peaks slightly below $R_{OO} = 2.42$ Å, the most probable excess-proton position at $d = 0$. Most paths, in fact, 77%, cross for $R_{OO} > 2.39$ Å, i.e., for values of $R_{OO}$ where a barrier along the $d$ coordinate is present. This means that the dominant mechanism for proton transfer is not one where the proton waits until the oxygen-oxygen separation $R_{OO}$ reaches small values so that the remaining barrier along $d$ is small or absent. Rather, protons cross the $d = 0$ midplane for a broad distribution of oxygen-oxygen separations $R_{OO}$ and by doing so overcome substantial free-energy barriers. This reverberates that a normal-mode analysis cannot account for all aspects of proton transfer in HCl solutions.

**Spectral signatures of proton transfer**. In order to dissect the excess-proton spectrum in Fig. 3c (black broken line) into contributions that have to do with proton-transfer events and those that do not, the excess-proton trajectories $d(t)$ are decomposed into three parts according to

$$d(t) = d_{TW}(t) + d_{TP}(t) + d_{NM}(t). \tag{2}$$

To illustrate this decomposition, Fig. 5a shows part of an example excess-proton trajectory, $d(t)$ (black line), together with the most likely excess-proton positions $d^*[R_{OO}(t)]$ (thin gray lines); the deviations between the black and gray lines visualize excess-proton motion relative to the oxygen it is bound to. We define the transfer-waiting contribution to the excess-proton trajectory as $d_{TW}(t) \equiv d^*[R_{OO}(t)]$ projected onto the closer branch of $d^*[R_{OO}(t)]$, shown as a blue solid line in Fig. 5b. Thereby, $d_{TW}(t)$ reflects the proton transfer jumps and also contains the water motion. The transfer-path contribution $d_{TP}(t)$ in Fig. 5c (red solid line) is defined as $d_{TP}(t) = d(t) - d_{TW}(t)$ during complete and incomplete transfer paths (as defined in Fig. 4e, g) and is zero elsewhere, it describes the excess-proton motion during transfer processes. Finally, by subtracting $d_{TW}(t)$ and $d_{TP}(t)$ from $d(t)$, we are left with the oscillations around $d^*[R_{OO}(t)]$ when the excess

proton is not undergoing a transfer, which constitutes the normal-mode contribution $d_{NM}(t)$ in Fig. 5d (green solid line). Different or more detailed excess-proton trajectory decompositions are certainly conceivable, the usefulness of the present scheme follows from its spectral decomposition properties.

In Fig. 5e the excess-proton spectrum, (black solid line) is decomposed as

$$\omega\chi'' = \omega\chi''_{TW} + \omega\chi''_{TP} + \omega\chi''_{NM}. \tag{3}$$

The power spectra of the transfer-waiting, $\chi''_{TW}$ (blue line), and transfer-path contributions, $\chi''_{TP}$ (red line), are computed from the $d_{TW}(t)$ and $d_{TP}(t)$ trajectories using the Wiener–Kintchine theorem (see "Methods" section for details). All cross-correlation contributions are included in the normal-mode contribution, $\chi''_{NM}$ (green line).

The normal-mode spectrum $\omega\chi''_{NM}$ in Fig. 5e accounts for the continuum band located between 2000 and 3000 cm$^{-1}$, it is in fact amenable to normal-mode analysis[19,25,27] but by construction does not include the proton-transfer dynamics. The range of the dominant normal-mode time scales included in $\omega\chi''_{NM}$, $\tau_{NM} = 11$–$17$ fs, follows from the spectral width of the continuum band, taken to be $f = 2000$–$3000$ cm$^{-1}$ in Fig. 5e, via $\tau_{NM} = 1/f$.

The transfer-path spectrum $\omega\chi''_{TP}$ in Fig. 5e shows a pronounced peak around 1200 cm$^{-1}$. An analytical model calculation shows that the peak in the transfer-path spectrum is related to the mean transfer-path time as $f_{TP} = 1/(2\tau_{TP})$[38]. Taking the results from the fits in Fig. 4f, yielding $\tau^c_{TP} = 14.1$ fs for complete and $\tau^i_{TP} = 12.7$ fs for incomplete transfer paths, we predict $f^c_{TP} = 1170$ cm$^{-1}$ and $f^i_{TP} = 1300$ cm$^{-1}$, indicated in Fig. 5e as vertical lines and which bracket the transfer-path peak very nicely.

The transfer-waiting spectrum $\omega\chi''_{TW}$ in Fig. 5e exhibits a peak around 400 cm$^{-1}$, a shoulder around 100 cm$^{-1}$ and a slow decay for lower frequencies. The peak around 400 cm$^{-1}$ (12 THz) is caused by oscillations of the oxygen–oxygen separation, $R_{OO}$, which couple to the proton position $d$ via the most likely proton position $d^*[R_{OO}(t)]$; in simple terms, the proton vibrates with the water molecule it is bound to. The oxygen vibrational time scale $\tau_{R_{OO}} = 86$ fs, indicated in Fig. 5b, follows from the peak of the power spectrum of $R_{OO}$ around 400 cm$^{-1}$, which is plotted in the lower panel of Fig. 5e and agrees perfectly with the peak in $\omega\chi''_{TW}$. This peak is in fact also well visible in our experimental THz/FTIR difference spectra, shown again in Fig. 6a as a broken red line for a 6 M HCl solution, the dotted red line shows the corresponding simulated difference spectrum. Note that this translational vibration of two water oxygens in the transient $H_5O_2^+$ complex is about twice as fast as the translational vibration of two hydrogen-bonded water molecules in pure water, which gives rise to the well-known IR signature around 200 cm$^{-1}$, shown as a blue solid line in Fig. 6b obtained from pure-water simulations[12]. This frequency shift is the reason why the water-vibration peak appears prominently in the difference spectra in Fig. 6a.

The shoulder in $\omega\chi''_{TW}$ around 100 cm$^{-1}$ is related to the transfer waiting time $\tau_{TW}$, which is the average time between two consecutive complete proton-transfer events, as predicted from an analytically solvable barrier-crossing model[38]. In Fig. 5f we show distributions of the transfer-waiting first-passage times, i.e., distributions of the time difference between crossing the most likely proton position $d^*[R_{OO}(t)]$ on one side of the midplane $d = 0$ and crossing $d^*[R_{OO}(t)]$ on the other side of the midplane $d = 0$ for the first time, for the three HCl concentrations. The distributions are essentially exponential in nature, which means that transfer events occur at a roughly constant rate and reflects the stochastic nature of the excess-proton transfer process in this

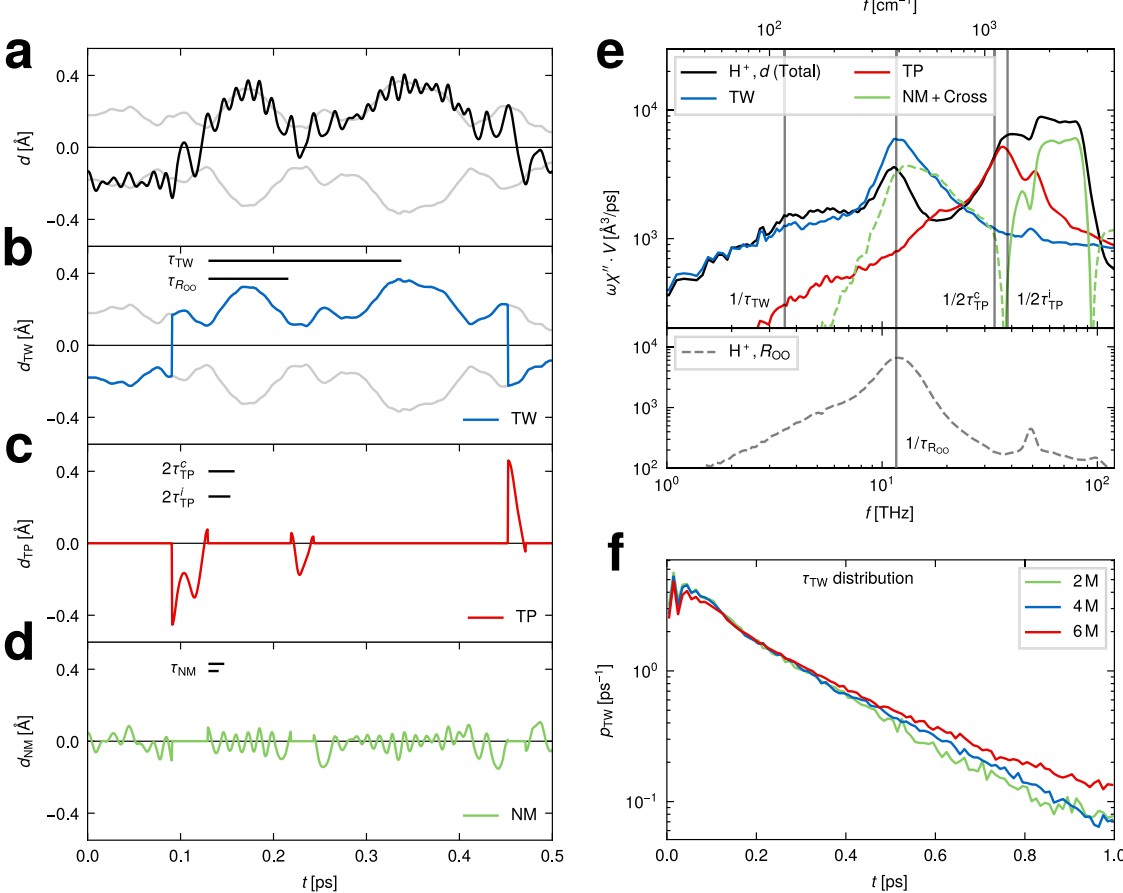

**Fig. 5 Spectral signatures of proton transfer. a–d** Decomposition of an excess-proton trajectory $d(t)$ (black solid line in **a**) into transfer-waiting $d_{TW}$ (blue solid line in **b**), transfer-path $d_{TP}$ (red solid line in **c**) and normal-mode contributions $d_{NM}$ (green solid line in **d**). The time course of the $R_{OO}$-dependent most likely proton positions at $d^*[R_{OO}(t)]$ are shown as thin gray solid lines in (**a** and **b**). **e** IR spectrum of the excess-proton motion projected onto $d$, shown as a black solid line, and IR spectra of the contributions shown in **a–c** in the respective color (the green broken line denotes negative values). The power spectrum of the $R_{OO}$ coordinate is shown as a gray broken line in the lower panel. The inverse of the characteristic time scales $\tau_{TW}$, $\tau_{R_{OO}}$, $2\tau_{TP}^c$ and $2\tau_{TP}^i$ are shown as thin vertical solid gray lines. **f** Distributions of the transfer-waiting first-passage times of excess protons, see main text for details.

reduced one-dimensional description. The mean of these first-passage distributions defines the transfer-waiting time $\tau_{TW}$, which is given in Table 1 and increases with rising HCl concentration. This indicates that hydronium ions have a slightly longer life time at higher HCl concentrations. In contrast, both complete and incomplete transfer-path times interestingly show no dependence on the HCl concentration. The inverse of the transfer-waiting time $\tau_{TW} = 280$ fs for 6 M, which is shown in Fig. 5e as a vertical line, is located at 120 cm$^{-1}$ (3.5 THz) and corresponds well to the position of the shoulder in $\omega\chi_{TW}''$, which confirms the connection between the transfer-waiting time and the spectroscopic signature around 100 cm$^{-1}$ that is predicted by analytical theory[38]. We note that the total length of the continuous proton-transfer trajectories are roughly twice as long as the mean transfer-waiting times, meaning that typically a few back-and-forth proton-transfer events occur in each trajectory (see Supplementary Note 7 for more details).

The characteristic time scales of each contribution, i.e., the transfer-waiting time $\tau_{TW} = 280$ fs, the water-oxygen vibrational time $\tau_{R_{OO}} = 86$ fs, the transfer-path times $\tau_{TP}^c = 14.1$ fs and $\tau_{TP}^i = 12.7$ fs and the normal-mode times $\tau_{NM} = 11$–17 fs are unambiguously extracted from the simulations and characterize both the trajectory contributions in the time domain in Fig. 5a–d, where they are included as horizontal black bars, and also the different spectral contributions in Fig. 5e.

We comment on the subtle spectral features in the range 1400–1800 cm$^{-1}$ in Fig. 5e, where small but distinct peaks are revealed in the different spectral contributions. The transfer-waiting contribution (blue line) peaks at about 1650 cm$^{-1}$, hinting to a weak coupling to an unperturbed water bending mode of the flanking water molecules. The transfer-path contribution (red line) peaks at 1750 cm$^{-1}$, the location of the experimental acid bend signature, which suggests that the acid bend couples particularly to the transfer path motion of the excess proton. Note, that even though the acid bend is primarily produced by the excess-proton motion orthogonal to the $d$ coordinate, this contribution to the isotropic spectrum is largely compensated by motion of the flanking water molecules, as shown in Supplementary Fig. 11. The normal-mode contribution (green line) peaks around 1500 cm$^{-1}$, consistent with previously calculated normal-mode spectra of Eigen-like solvated proton structures[25]. We thus see that our trajectory-decomposition technique also allows to disentangle the various normal-modes obtained for the solvated excess-proton complex.

So far we have concentrated on the excess-proton spectral contribution and not discussed the chloride contribution. The decomposition of the total simulated 6 M HCl spectrum in Fig. 6b (red solid line) into the chloride contribution (green solid line, including all cross correlations) and the remainder (gray solid line) demonstrates a prominent chloride peak around 150 cm$^{-1}$, which

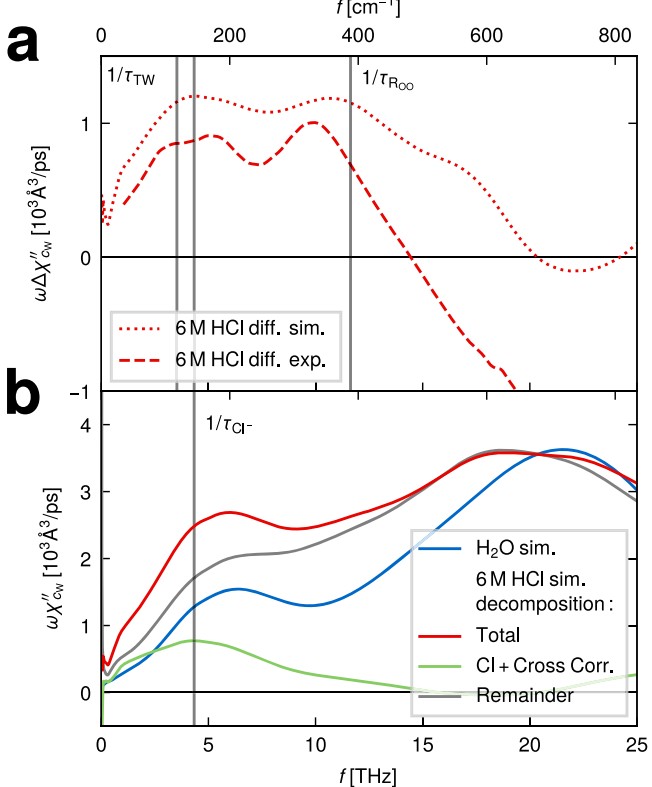

**Fig. 6 Absorption spectra in the THz regime. a** Experimental THz/Fourier-transform infrared (THz/FTIR) difference spectra of 6 M HCl solution (red broken line) compared to the difference spectrum from ab initio molecular dynamics (MD) simulations (red dotted line). **b** The simulated 6 M HCl spectrum (red solid line) is decomposed into a chloride-ion contribution (green solid line, including cross correlations) and the remainder (gray solid line). For comparison the simulated pure-water spectrum (blue solid line) is also shown.

**Table 1 Characteristic time scales of excess-proton transfer dynamics.**

| conc. | 2 M | 4 M | 6 M |
|---|---|---|---|
| $\tau_{TW}$ [fs] | 208 ± 6 | 229 ± 4 | 283 ± 4 |
| $\tau_{Cl^-}$ [fs] | 241 ± 3 | 249 ± 3 | 231 ± 3 |
| $\tau_{R_{OO}}$ [fs] | 81 ± 3 | 82 ± 2 | 86 ± 2 |
| $\tau_{TP}^c$ [fs] | 14.08 ± 0.08 | 14.34 ± 0.05 | 14.10 ± 0.04 |
| $\tau_{TP}^i$ [fs] | 12.65 ± 0.09 | 12.80 ± 0.05 | 12.74 ± 0.05 |
| $\tau_{NM}$ [fs] | (11 ± 1) — (17 ± 1) | | |

The standard errors of the transfer-waiting time $\tau_{TW}$ and the transfer-path times $\tau_{TP}^i$ and $\tau_{TP}^c$ are estimated from the variances of the fitted distributions. The errors of the normal-mode times $\tau_{NM}$, the oscillation time of the two flanking water molecules $\tau_{R_{OO}}$ and the rattling time of chloride ions $\tau_{Cl^-}$ are estimated from the resolution of the underlying spectra.

translates to a corresponding time scale of $\tau_{Cl^-} = 185$ ps and is due to the rattling of a chloride in its hydration cage[9,10]. This peak is also seen in the simulated difference spectrum in Fig. 6a (red dotted line) and is slightly shifted to larger frequencies in the experimental difference spectrum (red broken line). At 100–200 cm$^{-1}$ the remainder contribution in Fig. 6b (gray solid line) is significantly stronger than the pure-water spectrum (blue line), indicating that a process related to excess-proton motion significantly contributes in this wavenumber range. We suggest that this process is the excess-proton transfer-waiting contribution, which in Fig. 5e is shown to produce a broad shoulder around 100 cm$^{-1}$.

While the generally good agreement between our simulated and experimental spectra supports our chosen simulation methodology, it is clear that our classical treatment of nuclei motion is a drastic approximation and therefore some of the agreement might be due to fortuitous cancellation of errors. Interestingly, previous studies found no significant differences between IR spectra computed from simulations with and without nuclear quantum effects below 3000 cm$^{-1}$ [34,35,55], which might suggest that quantum-mechanical zero-point motion influences the excess-proton dynamics less than the instantaneous excess-proton distribution. We discuss quantum nuclear effects and basis set issues in the "Methods" section and Supplementary Note 1. Furthermore, we also compare our simulation results for radial distribution functions[28,33,52,56,57] (Supplementary Note 4) and proton diffusion coefficients[28,33,58–60] (Supplementary Note 8) with previous reports and discuss other observables that have been used in literature to characterize excess-proton transfer dynamics, such as identity correlation functions[27,58,61] (Supplementary Note 7), the hydrogen-bond asymmetry around hydronium ions[35] and the number of hydrogen bonds hydronium ions participate in[34,57,59] (Supplementary Note 9).

## Discussion

We show that the spectroscopic signature of proton-transfer dynamics between two water molecules in hydrochloric acid (HCl) solutions can be investigated by trajectory decomposition into transfer-waiting (characterized by the time scale $\tau_{TW}$), transfer-path (characterized by $\tau_{TP}$) and normal-mode contributions (characterized by $\tau_{NM}$). The decomposition is performed in the two-dimensional coordinate system that is spanned by the excess-proton position and the oxygen-oxygen distance of the two flanking water molecules and operates both in the time domain as well as in the frequency domain. The coupling of the excess-proton motion to the relative oscillations of the two flanking water molecules produces a fourth spectral proton-dynamics contribution (characterized by $\tau_{R_{OO}}$). The dynamics of each of the four contributions are described by distinct time scales with the ordering $\tau_{TW} > \tau_{R_{OO}} > \tau_{TP} > \tau_{NM}$ and therefore contribute with distinct peaks to the excess-proton IR spectrum. Due to the high proton charge, the excess-proton spectrum, and due to correlation effects with the neighboring water molecules in particular the excess-proton motion along the axis connecting the oxygens, contributes significantly to the IR difference spectrum of HCl. Our experimental THz/FTIR difference spectra resolve the slowest time scales, $\tau_{TW}$ and $\tau_{R_{OO}}$, of which the former one is overlaid by an additional spectral contribution due to rattling of the chloride ions, characterized by yet another time scale $\tau_{Cl^-}$, which is close to $\tau_{TW}$. Mid-IR experimental difference spectra from literature on the other hand are compatible with our predicted spectra associated with the $\tau_{TP}$ and $\tau_{NM}$ time scales.

In contrast to the transfer-path time $\tau_{TP}$, the transfer-waiting time $\tau_{TW}$ shows a weak dependence on the HCl concentration. Possible reasons include ionic screening and repulsion effects between neighboring excess protons, but also entropic effects due to the reduced number of accepting water molecules have been discussed[28,32,62]. This is consistent with experimental results showing a decrease of the excess-proton diffusivity with increasing HCl concentration[60], which is reproduced in our simulations (see Supplementary Fig. 19). One should note that the transfer-waiting times in local transient $H_5O_2^+$ complexes include back-and-forth proton-transfer events, which do not contribute to the long-time excess-proton diffusion[28,33,58,59,63] but nevertheless have a pronounced spectroscopic signature[38].

Our results nicely complement recent normal-mode calculations. We find that the continuum band stems from normal-

mode vibrations of less symmetric, i.e., more Eigen-like, configurations of the excess proton, which are strongly influenced by the oxygen-oxygen separation $R_{OO}$[14,19,25,26]. On the other hand, our transfer-path signature, which is dominated by a broad absorption around 1200 cm$^{-1}$, shows striking similarity with normal-mode spectra computed for more symmetric, i.e., more Zundel-like, configurations of the excess proton[14,19,25,26]. It is not implausible that normal modes for small separations of the two flanking water molecules show similar spectroscopic signatures as the transfer paths we extract from our simulation trajectories. Yet, in a normal-mode picture the interconversion between the metastable Zundel-like and Eigen-like excess-proton states cannot be explained consistently, even though the importance of this process for a complete description of the mid-IR signatures was acknowledged several times[14,24–26,35]. In fact, the broad distribution of this interconversion time scale is demonstrated by the transfer-waiting time distribution and also gives rise to a distinct spectral signature, that we identify in the THz regime.

A recent study employing similar simulation techniques has decomposed the proton power spectra with respect to the proton asymmetry coordinate and thereby reached similar conclusions to ours: Eigen-like configurations give rise to the continuum band while Zundel-like configurations dominantly contribute around 1200 cm$^{-1}$ [47]. That study also determined the proton-transfer time scale using two-dimensional transition state theory and Marcus theory of ion pairing and finds this time scale to be concentration dependent, in agreement with our and previous observations.

In summary, in many theoretical treatments, only normal modes of meta-stable or stable states are assumed to produce spectral contributions. Any spectral mode is therefore interpreted as being due to a meta-stable state, consequently, broad spectral modes are often interpreted as reflecting a wide collection of normal modes with slightly different frequencies. In this paper, we show that transfer and barrier-crossing events of charged particles as well as transfer paths create spectral features by a mechanism that is very different from a normal-mode picture and that these spectral features are broadened by the stochastic nature of the transfer dynamics[38]. In fact, the strength or frequency of a spectral feature does not allow to tell whether it is caused by normal-mode oscillations in a stable or meta-stable state or whether it is caused by transfer or barrier-crossing dynamics.

## Methods

**Computational methods: ab initio molecular dynamics simulations.** The Born–Oppenheimer ab initio MD simulations of pure water and HCl solutions at three different concentrations were performed with the CP2K 7.1 software package using a polarizable double-zeta basis set for the valence electrons, optimized for small molecules and short ranges (DZVP-MOLOPT-SR-GTH, with the exception of the chloride anions, that were modeled including diffuse functions in the aug-DZVP-GTH basis set), dual-space pseudopotentials, the BLYP exchange-correlation functional, and D3 dispersion corrections[64–68]. The cutoff for the plane-wave representation was 400 Ry. The system parameters are summarized in Table 2.

Before production, each system was equilibrated in classical MD simulations for 200 ps under NPT conditions at atmospheric pressure and 800 ps under NVT conditions at 300 K, using the GROMACS 2020.5 software[69] with the SPC/E water

model[70]. The force fields for Cl$^-$ and H$_3$O$^+$ were taken from[71]. The ab initio MD simulations were subsequently performed using a time step of 0.5 fs under NVT conditions at 300 K by coupling the system to a CSVR thermostat with a time constant of 100 fs[72].

Dipole moments were obtained after Wannier-center localization of the electron density at a time resolution of 2 fs or 4 fs. At each time step, the Wannier centers were assigned to the closest oxygen or chloride ion. Water molecules were assembled by assigning each proton to the closest oxygen nucleus, thereby forming either water or hydronium ions. For the hydronium ions, all protons were treated as excess-proton candidates and further processed based on a dynamical criterion as discussed in the main text and Supplementary Methods 2. The dipole moments $\mathbf{p}$ follow as a sum over the respective position vectors $\mathbf{r}_i$ and charges $q_i$ ($q = 2$ e for Wannier centers, and reduced core charges for nuclei), $\mathbf{p} = \sum_i q_i \mathbf{r}_i$ for the whole or desired sub systems.

Linear response theory relates the dielectric susceptibility $\chi(t)$ to the equilibrium autocorrelation of the dipole moment $C(t) = \sum_D \langle \mathbf{p}(t)\mathbf{p}(0)\rangle$, reading in Fourier space

$$\chi(\omega) = \frac{1}{V k_B T \epsilon_0 D}\left(C(0) - i\frac{\omega}{2}\widetilde{C}^+(\omega)\right), \qquad (4)$$

with system volume $V$, thermal energy $k_B T$, vacuum permittivity $\epsilon_0$ and $D$ being the number of Cartesian dimensions of the polarization vector $\mathbf{p}$. IR spectra can therefore be calculated straight-forwardly from sufficiently sampled trajectories of the ab initio MD simulation data using Eq. (4) and the Wiener–Kintchine relation, derived in Supplementary Methods 4 as

$$C(t) = \frac{1}{2\pi(L_t - t)}\int_{-\infty}^{\infty} d\omega\, e^{-i\omega t}\,\tilde{\mathbf{p}}(\omega)\tilde{\mathbf{p}}^*(\omega), \qquad (5)$$

where $\tilde{p}(\omega)$ is the Fourier-transformed dipole-moment trajectory with length $L_t$ and the asterisk denotes the complex conjugate. Alternatively, for charged subsystems, as in the case of the chloride ions, the computation using the time derivative of the polarization, i.e., the current $\mathbf{j} = \frac{d}{dt}\mathbf{p}(t)$, is preferable

$$C(t) = \frac{1}{2\pi(L_t - t)}\int_{-\infty}^{\infty} d\omega\, \frac{e^{-i\omega t}}{\omega^2}\,\tilde{\mathbf{j}}(\omega)\tilde{\mathbf{j}}^*(\omega). \qquad (6)$$

Quantum corrections have previously been addressed[73], but were not applied here.

Since the Wannier-center localization time step $\Delta t_{WC} = 4$ fs is larger than the original simulation time step $\Delta t = 0.5$ fs, the analysis is performed on two types of trajectories stemming from the same simulations: one set of trajectories containing the electronic degrees of freedom and another set of trajectories of higher time resolution but only containing nuclei positions. This higher resolution data is used for the calculation of excess-proton spectra and kinetics.

All spectra were smoothed by a convolution with a Gaussian kernel of varying width, depending on their respective resolution. We used a standard deviation of 55, 20, and 50 cm$^{-1}$ for bulk spectra, the H$_5$O$_2^+$ complex difference spectrum in Fig. 3c and excess-proton spectra, respectively. Experimental data was smoothed using a standard deviation of 3 cm$^{-1}$.

To address the quality of the chosen basis set, shorter simulations at 6 M were performed using the non-short range basis set (DZVP-MOLOPT-GTH) as well as a triple-zeta doubly polarizable (TZV2P-GTH) basis set. Spatial correlations in the data are compared in Supplementary Fig. 9. While the coordination of excess protons with chloride ions slightly increases when the more elaborate basis sets are used, no significant differences in correlations between excess protons and oxygen nuclei are found, which are the focus of this study.

**Experimental methods: THz absorption measurements.** THz spectroscopic measurements in the 30–650 cm$^{-1}$ frequency range were done with a commercial Fourier Transform spectrometer (Bruker Vertex 80v, Germany) equipped with a mercury light source and a liquid helium cooled bolometer detector (Infrared Laboratories, Germany). Spectra result from an average of 128 scans with a resolution of 2 cm$^{-1}$. The liquid sample cell is composed of diamond windows (Diamond Materials GmbH, Germany) in which a Kapton spacer of approximately 13 μm was placed between the windows to fix the sample thickness. The exact thickness of the sample cell was determined from the etaloning pattern of the empty sample cell. The temperature of the sample was held constant at 20.0 ± 0. 2 °C by an external chiller. The measured frequency-dependent extinction coefficient, $\alpha_{\text{solution}}(\omega)$, is determined using the Beer–Lamber law

$$\alpha_{\text{solution}}(\omega) = \frac{1}{d}\ln\left(\frac{I_{\text{water}}(\omega)}{I_{\text{solution}}(\omega)}\right) + \alpha_{\text{water}}(\omega), \qquad (7)$$

where $d$ is the sample thickness, $I_{\text{water}}(\omega)$ and $I_{\text{solution}}(\omega)$ are the experimental transmitted intensities of the water reference and the sample. $\alpha_{\text{water}}(\omega)$ is the extinction coefficient of bulk water and is taken from literature[74]. The extinction coefficient $\alpha(\omega)$ is converted to the absorption spectrum, proportional to the imaginary part of the dielectric susceptibility $\chi''(\omega)$, by fitting the spectra and performing a Kramers–Kronig transform as presented in Supplementary Methods 1.

**Table 2 Parameters of the ab initio molecular dynamics simulations.**

| Conc. | 0 M | 2 M | 4 M | 6 M |
|---|---|---|---|---|
| $V^{\frac{1}{3}}$ | 19.73 Å | 20.25 Å | 20.25 Å | 20.23 Å |
| $N_{H_2O}$ | 256 | 258 | 244 | 224 |
| $N_{H^+}, N_{Cl^-}$ | 0 | 10 | 20 | 30 |
| $\tau$ | 201 ps | 52 ps | 84 ps | 84 ps |
| $\Delta t_{WC}$ | 2 fs | 4 fs | | |

The difference absorption spectra of HCl solutions with respect to pure water and normalized with respect to the water concentration are given by

$$\omega\Delta\chi''_{c_W}(\omega) = \frac{1}{c_W}\omega\chi''_{\text{solution}}(\omega) - \frac{1}{c_W^0}\omega\chi''_{\text{water}}(\omega), \tag{8}$$

where $c_W$ and $c_W^0$ are the concentration of water in the aqueous HCl solutions and bulk water, respectively, determined from the solution density at room temperature.

## Data availability

The datasets generated and analyzed during the current study are available from the corresponding author on request.

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

## Acknowledgements

We gratefully acknowledge computing time on the HPC clusters at the physics department and ZEDAT, FU Berlin, as well as the computational resources provided by the North-German Supercomputing Alliance (HLRN) under project bep00068 (R.R.N.). This work was funded by the Deutsche Forschungsgemeinschaft (DFG) under Germany's Excellence Strategy - EXC 2033 - 390677874 - RESOLV (M.H.) and via grants SFB 1078, project C1 (R.R.N.) and SFB 1349, project C4 (R.R.N.). Funding by the European Research Council (ERC) is acknowledged via Advanced Grant 695437 THz Calorimetry (M.H.) and Advanced Grant 835117 NoMaMemo (R.R.N.).

## Author contributions

F.N.B. and R.R.N. conceived the theory and designed the simulations. F.N.B. performed the ab intio MD simulations. F.N.B. and M.R. analyzed the data and designed the figures. E.A. and M.H. carried out the THz measurements and analyzed the data. All authors discussed the results, analyses, and interpretations. F.N.B., M.H., and R.R.N. wrote the paper with input from all authors.

## Funding

## Competing interests

The authors declare no competing interests.
