## [Peer Review File · Nature Communications]

REVIEWER COMMENTS

Reviewer #1 (Remarks to the Author):

In this study, the authors performed ab initio molecular dynamic simulations and THz dielectric spectroscopy on aqueous hydrochloric (HCl) acid solutions at several different concentrations. The main focus of this study is the decomposition of the infrared difference spectra, in which several characteristic regions are identified. The characteristic motions are assigned by the help of simulations. The authors attributed 1800-3000 cm^{-1} to the normal mode motions of proton-water complex, $\sim 1200 \text{ cm}^{-1}$ to the barrier crossing, and $\sim 100 \text{ cm}^{-1}$ to the waiting. Also, a 400 cm^{-1} band “is shown to be caused by the coupling of the proton motion to the relative oscillations of the two flanking water molecules in transient H_5O_2^+ complexes”, which may be the rattling of the proton between the two adjacent oxygen atoms, if my understanding is correct. Based on the above identifications of the characteristic motions of proton, the authors propose that the proton-transfer (PT) motion is stochastic in nature.

I am not convinced by the stochastic nature of PT, and thus do not support the publication of the manuscript in the current form. On the contrary, I expected some discussions on a decisive mechanism, at least in attempt, on (1) what triggers a transfer event? And (2) as each of the the three hydrogens on a hydronium can potentially be the “excess” proton, what decides which proton actually becomes the “excess” one? The current study seems to treat PT as isolated event that involves only two or three waters, and this may be the reason that the authors support the stochastic nature of PT.

I have some further comments below.

1. It would be nice if the authors could report the diffusion coefficient of proton in order to give the reader a sense on the accuracy of the simulation.

2. The acid concentrations in this study is quite high. Thus, it may be difficult to ignore the influence of a neighboring proton and/or a chloride anion. The authors seems to treat PT in such case as independent events. However, considering a 2M HCl aqueous solution, there are on average 23 waters surrounding an H — Cl pair, assuming an even distribution of the acid molecules. In such case, the correlation of the co-ion and counter-ion needs to be taken into account.

3. A PT event in solution must be accompanied by hydrogen bond breakage/formation beyond the second hydration shell, and thus involve the correlated motions of many waters and, possibly, the other ions. It would be nice if the authors could interpret such correlated motion from simulation and spectroscopy, if it is not impossible.

4. The distribution of R_{OO} of crossing in Figure 4D may not be fair, because the most probable distribution of R_{OO} must be peaked at $R_{OO} > 2.42 \text{ \AA}$, regardless of PT. Thus, it may be of more interest to compare the probability of crossing at R_{OO} weighted by the total probability at R_{OO} . Also, non-barrier crossing event is often observed in molecular dynamics, from which the 2D (d, R_{OO}) free energy map is statistically averaged. I do not see the reason to use this analysis, or the 2D ($d, R_{RR_{OO}}$) free energy map, to support stochastic model, which is kinetic in nature. By the way, I find the successive shoulders in Figure 4D may be of interest because they may favor PT event at these specific O—O separations.

5. It is of interest to note from Figure 3c that all the features of the difference spectra in the bulk also appear in the difference spectrum between Zundel and two waters. To the reviewer, such observation seems to indicate that none of the bands in the infrared difference spectra corresponds to the PT event, because such event does not actually exist in Zundel. On the other hand, the characteristic band of such event may be buried in the signal of proton motions, either bonded or excess.

Reviewer #2 (Remarks to the Author):

I have read the article entitled, "Spectral signatures of stochastic proton transfer events in aqueous hydrochloric acid solutions" with great interest. This is a beautiful piece of theoretical/experimental work. This is my issue with this manuscript. This sentence, "While the stochastic time scales of aqueous proton-transfer events have been studied recently [36], their identification in experimental spectra is a main result of this work." There is then no comparison with the work of ref [36] in accounting for the time scales and mechanisms. Ref [36] appears to be doing something similar--but this paper clearly goes beyond this reference. The question to this reviewer is if this work supersedes the work of Ref [36] or builds on it. This is important to clarify. This manuscript is certainly well done and has incredible analysis. But what is missing is a comparison to other high-quality science (cited, but not discussed). This topic is important and I believe with such a high profile article, the onus is on these authors to put their work in the proper context. Perhaps a statement on what is in agreement with [36] and what is in disagreement with [36].

Please comment on the work of <https://pubs.acs.org/doi/10.1021/jp501091h> (not cited) needs to be brought to light. The finding of a correlated structure in acids is important. To the extent that the counter-ion is not a Spector should be captured (somehow) in this study. This goes well beyond HCl and will be important for other acids (such as the oxyanions) that also show this correlated behavior.

I urge the authors to put this excellent study work in proper context. This is important because the same code was used (CP2K) and a similar reactive picture was used. This work is clearly novel and goes beyond the original experimental work of Tokmakoff (ref 14) and corroborates and provides a reinterpretation of many of the details in his pioneering study. The new experiments are exciting. Having some consistency between the theoretical concepts/simulations would be useful for the community.

If the authors can put their work in proper context of published work in concentrated acids, then this work will be an excellent contribution to the Nature Communications.

Reviewer #3 (Remarks to the Author):

The manuscript "Spectral signatures of stochastic proton transfer events in aqueous hydrochloric acid" by Brünig et al. presents the results of a joint experimental and computational investigation of aqueous solutions of hydrochloric acid and their vibrational spectra. In my opinion, this manuscript is unfortunately not suitable for publication in Nature Communications. After some modifications, it could be suitable for publication elsewhere. Below, I explain my reasons for this conclusion.

A large part of the work focuses on the molecular dynamics simulations and on the analysis of the resulting trajectories - the identification of the excess protons and assignment of spectral features. These molecular dynamics simulations are performed with a classical, rather than quantum, description of the nuclei, and one particular GGA functional with a dispersion correction. The results are then extensively analyzed and the identified spectral features are matched to observed experimental features.

My major concern is that for an acid solution, classical nuclei are a major approximation that is not appropriate for many properties, spectroscopic or others, of these systems. In simulations that do include nuclear quantum effects, results would be substantially different from those presented in this paper. In that case, much of the very detailed analysis would have to be performed again, or in

fact re-designed so that the quantum delocalization of the nuclei could be included in a meaningful way. It is not clear at this point which, if any, of the results and their interpretation in the presented manuscript would remain valid if this was done. As the manuscript stands, nuclear quantum effects are mentioned once in the Results section:

"While simulations including quantum-mechanical treatment of the nuclei may be the more accurate model [38, 39], the computationally cheaper Born-Oppenheimer approach was taken in favor of improved statistics for the stochastic analysis."

This seems to constitute a considerable logical error - it is in principle possible that even if a description of nuclear quantum effects would be prohibitively expensive, it would still be crucial to say anything reliable about the system of interest. It is up to the authors to convince the reader that they can draw useful conclusions from their classical simulations for this system for which the importance of nuclear quantum effects is well known. The authors have not even attempted to convince the reader that this is the case. A manuscript that does make such an attempt would have to be considered again, but it seems this would mean major changes throughout the work. I also note that the distinction between a "Born-Oppenheimer approach" and "quantum-mechanical treatment of the nuclei" in the above quoted sentence is wrong. Nuclei can be treated quantum mechanically under the Born-Oppenheimer approximation, and, in fact, that is exactly what imaginary time path integral simulations do - quite a common approach to including NQEs in aqueous solutions. This makes me worry if the authors appreciate what kind of quantum effects come into play in their system and how.

NQEs are mentioned again in Conclusions, noting that including them stabilizes Zundel configurations compared to Eigen. I find it very strange that the authors would say that their main result is not dramatically modified by this. Their analysis is centered around the characteristic V-shaped effective potential for the excess proton, and if this were to turn into a single-well potential with quantum delocalization, it is not clear, what aspects, if any, of the analysis would be applicable and if the conclusions would be the same. This would certainly have to be shown explicitly, rather than asserted at the very end of the manuscript. The authors even cite a published paper that does include NQEs and uses projection to interpret vibrational spectra, but they do not try to compare to this work.

As such, I find the way the authors approach their neglect of nuclear quantum effects unconvincing and consider it a major issue with the current manuscript.

Even with classical nuclei, some of the analysis relies on specific features of the shape of the effective potential. If this were to change with a different density functional, would the analysis

remain the same? This is unclear from the text. It is known in the literature that the combination of classical nuclei and GGA functionals results in a cancellation of errors that yields surprisingly good results for some quantities, compared to experiment. Regardless of how this cancellation still does or does not apply to systems with proton defects, a good match to experiment at this level of methodology should be viewed with a healthy degree of skepticism. The authors take the results they get with their particular functional and classical nuclei at face value note a good match with experiment.

The projection analysis of vibrational spectra itself is potentially interesting and helpful for interpreting experiments, if it could be presented in a way that is general for any proton defect, rather than relying on the existence of specific minima or barriers in the effective potential for the excess proton. One thing I am missing in this regard is an explanation of how this effective potential varies with the changing broader surrounding of the defect in the solution, again something that has been discussed in the literature before. This is important when talking about specific trajectories, such as the ones shown as the various examples, because these most likely do not evolve in the average potential, but rather an instantaneous distortion of it. If these distortions evolve on a time scale similar to or slower than the proton transfer trajectories analyzed here (and from the literature that appears to be the case), talking about the trajectory crossing specific barriers or other landmarks in the average potential might be tricky.

The distinction between Eigen and Zundel configurations based on the d coordinate (called δ or ν in the existing literature, by the way) and saying that Zundel has $d=0$ seems very simplistic and not useful.

The electronic structure setup seems reasonable, though some benchmarking of the short-range double-zeta molopt basis might be warranted, perhaps compared to the non-short-range double-zeta molopt and non-molopt triple-zeta basis sets.

There are other, smaller and more specific, comments that could be made on this paper, but that seems premature until the above major concerns are addressed.

Reviewer #4 (Remarks to the Author):

In this manuscript, the authors use ab initio molecular dynamics to compute the IR spectrum of the aqueous proton from a dynamical perspective instead of typical normal mode analyses. Despite

decades of experimental and theoretical efforts, the physical, dynamical, and spectroscopic properties of the aqueous proton remain incredibly challenging to pin down. There has been a recent resurgence in this area thanks to rapid advancements in experimental (particularly ultrafast IR) and computational methods, but many questions remain on the interpretation of the IR and 2D IR spectra of the aqueous proton. The dynamical perspective presented in in this manuscript is a novel contribution to the many studies on the aqueous proton.

The manuscript as presented, in my opinion, requires some major revisions before I can fully endorse publication. My comments/suggestions/concerns are presented below:

1. I think there is an excellent opportunity to discuss and present the dynamical perspective given here with both the normal mode and experimental interpretations recently published by Tokmakoff and Bowman (JPCB, 2019, 123, 7214 and JCP, 2020, 153, 124506) which I believe is the current standard for our understanding of the aqueous proton. Somehow these papers are not even cited.

I do not support the dismissive attitude towards normal mode approaches, especially the sentence in the Introduction, “normal-mode approximations, which however fail, for an obvious reasons”. I agree that there are issues with the normal mode perspective, but I see the normal mode perspective and this new dynamical perspective as highly complementary and I think they should be presented as such. In fact, the normal mode perspectives of Tokmakoff and Bowman have been quite successful in interpreting many of the features in the IR and, more impressively, 2D IR spectra. For example, the “superharmonic” behavior of the shared proton stretch at 1200 cm⁻¹ and the bending motion of the aqueous proton complex near 1750 cm⁻¹, which is highly coupled and mixed with the proton stretch.

2. Along this line, the current paper doesn't address the acid bending mode at 1750 cm⁻¹. A feature in this region isn't clearly reproduced by the simulations, perhaps at higher concentrations. Since the proton stretch and bend are highly coupled and mixed, I would anticipate the “transition path” dynamics that give rise to the 1200 cm⁻¹ feature should also contribute to the acid bending region.

3. I have some questions/concerns regarding the definition of “proton transfer events”. The authors state “The actual proton transfer involves the thermally activated crossing of an energy barrier...” I agree with this statement, but the timescales of the “barrier crossing” event do not match up with experiments. Polarization anisotropy measurements by Tokmakoff (JPCB, 2018, 122, 2792 and JCP, 2019, 151, 034501) showed a slow 2.5 ps decay and was interpreted in terms of irreversible proton transfer (not just rattling) that requires overall reorganization of the local H-bond network, in agreement with NMR measurements and known H-bond switching timescales in water. This is the barrier that must be crossed for proton transfer.

In my opinion, the fast dynamics captured in these simulations are not irreversible Grotthuss proton transfer events but rather fast rattling of the proton within the proton complex (either between a special pair in a Zundel picture or Greg Voth special pair dance in an Eigen picture). The proton stretch 2D IR spectra of Elsaesser and Tokmakoff are consistent with a low-barrier proton stretch potential where the zero-point level lies above the barrier between proton donor and acceptor. Therefore, the proton wavefunction is highly delocalized and hence the interpretation in terms of more Zundel-like configurations.

2D IR supports a relatively localized and persistent Zundel-like complex despite fast H-bond fluctuations. How do these recent experimental results/interpretations fit into the presented simulations? On what timescales are these simulations capturing irreversible transfer and H bond rearrangements? Are these simulations really only tracking fast rattling/special pair dancing? If so, I would not use the phrase "proton transfer" but rather "proton dynamics".

4. The authors admit that their simulations support a more Eigen-centric picture, in contrast to the experimental interpretations discussed above, but state that an Eigen or Zundel picture wouldn't change the results. This makes me concerned that the simulations are getting the right answers (or at least reasonable agreement with experiment) for the wrong reasons.

Tokmakoff and Bowman have been careful in their definition of Eigen and Zundel. Although there is a large distribution of O-O and O-H distances that span traditional Eigen and Zundel definitions, the 2D IR spectra support a more localized special pair arrangement, i.e., and strongly bound proton between two unique, special water molecules. Hence, the description of the complex as "Zundel-like". Using these more relaxed definitions, do the simulations presented here support this view, or are they really more Eigen-centric? The presented PES's indeed seem more Eigen centric and the large barrier to proton shuttling between nearest neighbors is a bit concerning given the current view of the aqueous proton.

5. One place where the normal mode picture does fail is in the continuum region near 2500 cm^{-1} . 2D IR spectra show all the spectral signatures of the aqueous proton being highly coupled, pointing to a single and indistinguishable Zundel-like proton complex. Normal mode analyses predict this region to arise from more Eigen-like OH stretches and predict weak coupling to the proton stretch and acid bend modes. Tokmakoff and Bowman suggest this difference could be from fast fluctuations of the complex that can't be captured in the normal mode approach. Can the authors provide any more insight into proton stretch-bend-continuum coupling from the dynamical perspective that could help interpret the experimental 2D IR observations?

In summary, I would suggest that the authors discuss their results in the context of the normal mode and experimental analyses of Bowman, Tokmakoff, and others, and present their results as complementary and as providing a new perspective. What new insights do these simulations give us? Do they change how we should think about the aqueous proton? Do they offer new or alternative interpretations of experimental data compared to normal mode approaches? Are any new interpretations provided that can't be answered by normal mode approaches?

Prof. Dr. Roland Netz
Arnimallee 14
14195 Berlin**Telefon** +49 30 838-55737
Fax +49 30 838-53741
E-Mail rnetz@physik.fu-berlin.de
Internet www.physik.fu-berlin.de/
einrichtungen/ag/ag-netz

13.04.2022

Manuscript "Spectral signatures of excess-proton waiting and transfer-path dynamics in aqueous hydrochloric acid solutions"

To the referees,

We appreciate the thoughtful reports by the referees. In the following, we reproduce the referees' comments in full and explain the changes made to our paper. Apart from minor changes, all additions to our paper are in the marked-up version shown in red. We would like to remark that beyond their constructive criticism, the referees appreciate our work as "a beautiful piece of theoretical/experimental work" with "incredible analysis" (reviewer 2) and note that "the dynamical perspective presented in in this manuscript is a novel contribution" (reviewer 4) and "the projection analysis of vibrational spectra itself is potentially interesting" (reviewer 3). We revised and expanded our manuscript considerably and hope that the referees find the new version of our paper publishable in Nature Communications.

Reviewer #1 (Remarks to the Author):

In this study, the authors performed ab initio molecular dynamic simulations and THz dielectric spectroscopy on aqueous hydrochloric (HCl) acid solutions at several different concentrations. The main focus of this study is the decomposition of the infrared difference spectra, in which several characteristic regions are identified. The characteristic motions are assigned by the help of simulations. The authors attributed $1800\text{-}3000\text{ cm}^{-1}$ to the normal mode motions of proton-water complex, $\sim 1200\text{ cm}^{-1}$ to the barrier crossing, and $\sim 100\text{ cm}^{-1}$ to the waiting. Also, a 400 cm^{-1} band "is shown to be caused by the coupling of the proton motion to the relative oscillations of the two flanking water molecules in transient H_5O_2^+ complexes", which may be the rattling of the proton between the two adjacent oxygen atoms, if my understanding is correct. Based on the above identifications of the characteristic motions of proton, the authors propose that the proton-transfer (PT) motion is stochastic in nature.

I am not convinced by the stochastic nature of PT, and thus do not support the publication of the manuscript in the current form. On the contrary, I expected some discussions on a decisive mechanism, at least in attempt, on (1) what triggers a transfer event? And (2) as each of the the three hydrogens on a hydronium can potentially be the "excess"proton, what decides which

proton actually becomes the “excess” one? The current study seems to treat PT as isolated event that involves only two or three waters, and this may be the reason that the authors support the stochastic nature of PT.

We thank the reviewer for these comments. There is no doubt that the excess-proton motion in our simulations is deterministic and for the main results of our paper, which are based on the decomposition of continuous excess-proton trajectories, we actually do not have to assume that proton transfer is stochastic in nature. However, the single-exponential distribution of transfer-waiting times in Fig. 5f in our manuscript shows that a stochastic description of proton transfer, according to which transfer events occur at constant rate independent of history, is possible and useful. This is the reason why we discuss proton-transfer events using a stochastic description in terms of transfer paths and transfer waiting-time distributions. To reflect that the stochastic nature of proton-transfer events is not a necessary assumption to obtain our main results, we have removed the word “stochastic” from the title and other places in the paper and stress that our spectral decomposition is based on a deterministic decomposition of trajectories.

Changes in the proton solvation structure that precede and in some sense cause proton transfer were investigated in literature before. For example, the structure of the local hydrogen-bond network around protons [1] and the role of a fourth water molecule that is hydrogen-bonded to the hydronium ion [2-4] have been shown to be good indicators of an enhanced probability for proton transfer. We have added section XI to the SI where we analyze our simulation data using these indicators and confirm the usefulness of these concepts, we added a short discussion in the Introduction as well as in a newly added subsection just before the Conclusions.

We stress that our paper is not about what causes proton transfer events, which has been studied in literature before, but rather how to dissect the spectrum of HCl solutions into contributions from different trajectory segments and in particular what the spectroscopic signature of proton transfer over a free-energetic barrier is. The latter contribution cannot be described in terms of a normal-mode picture and therefore poses a theoretically demanding problem.

More generally, we remark that the emergence of stochasticity in liquid many-body systems is a general phenomenon that can be elegantly studied by projection theory [5]. A rare event, such as a proton transfer over a barrier, could in principle be a phenomenon with a well-defined frequency, like a collective oscillation; for HCl solutions, however, we find the proton transfer to occur with an extremely broad and roughly exponential waiting-time distribution, see Fig. 5f in our manuscript. We added a short note on the exponential nature of our waiting-time distribution when we first discuss them. While a perturbed solvation structure of the excess proton can be used to predict a proton transfer event, one could ask what the cause of a perturbed solvation structure is, which would lead to an infinite regress. This is exactly the reason, why a stochastic description in terms of a reduced reaction coordinate is in literature used to describe barrier-crossing phenomena in overdamped many-body systems.

[1] Napoli, J. A., Marsalek, O. & Markland, T. E. Decoding the spectroscopic features and time scales of aqueous proton defects. *J. Chem. Phys.* 148, 222833 (2018).

[2] Fischer, S. A. & Gunlycke, D. Analysis of Correlated Dynamics in the Grothuss Mechanism of Proton Diffusion. *J. Phys. Chem. B* 123, 5536–5544 (2019).

[3] Tse, Y. L. S., Knight, C. & Voth, G. A. An analysis of hydrated proton diffusion in ab initio molecular dynamics. *J. Chem. Phys.* 142, 014104 (2015).

[4] Biswas, R., Tse, Y. L. S., Tokmakoff, A. & Voth, G. A. Role of Presolvation and Anharmonicity in Aqueous Phase Hydrated Proton Solvation and Transport. *J. Phys. Chem. B* 120, 1793–1804 (2016).

[5] Ayaz C., Dalton B. A. & Netz R. R., Generalized Langevin Equation with a Non-Linear Potential of Mean Force and Non-Linear Memory Friction From a Hybrid Projection Scheme, arXiv:2202.01922 (2022),

I have some further comments below.

1. It would be nice if the authors could report the diffusion coefficient of proton in order to give the reader a sense on the accuracy of the simulation.

We thank the reviewer for this comment. We have analyzed the diffusion properties of excess protons and oxygens from our simulations and find good agreement with previous simulation studies. There is also good agreement with experimental data for the ratio of the excess proton and the oxygen diffusivity. All results are discussed in section X of the SI and briefly mentioned in the Introduction as well as in a newly added subsection just before the Conclusions.

2. The acid concentrations in this study is quite high. Thus, it may be difficult to ignore the influence of a neighboring proton and/or a chloride anion. The authors seems to treat PT in such case as independent events. However, considering a 2M HCl aqueous solution, there are on average 23 waters surrounding an H — Cl pair, assuming an even distribution of the acid molecules. In such case, the correlation of the co-ion and counter-ion needs to be taken into account.

We thank the reviewer for this comment. In the newly added SI section XX, we show that even for the highest acid concentration of 6M, only 5% of all excess protons have a chloride ion instead of a water molecule as a second nearest neighbor. There we also show that no significant spatial correlations are found between hydronium ions. Correlations between excess protons and chloride ions, that were the focus of a previous study [1], are in fact observed at much higher concentrations around 11 M. We comment on this in the new version of our manuscript.

That the proton spectral signatures are not influenced by collective proton-proton or proton-chloride effects for concentrations up to 6M, is convincingly demonstrated by the fact that difference spectra of HCl solutions scale linearly in HCl concentration, which holds true in our simulations (see Fig. 1d in our manuscript) as well as in our experiments (see Fig. 2c in our

manuscript). We added some comments along these lines on pages 4 and 5 of the revised version of our manuscript.

[1] Baer, M. D., Fulton, J. L., Balasubramanian, M., Schenter, G. K. & Mundy, C. J. Persistent ion pairing in aqueous hydrochloric acid. *J. Phys. Chem. B* 118, 7211–7220 (2014).

3. A PT event in solution must be accompanied by hydrogen bond breakage/formation beyond the second hydration shell, and thus involve the correlated motions of many waters and, possibly, the other ions. It would be nice if the authors could interpret such correlated motion from simulation and spectroscopy, if it is not impossible.

We thank the reviewer for this comment. We have analyzed the role of a fourth water molecule as well as the local hydrogen-bond network with respect to proton-transfer events and report our findings in the newly added section XI in the SI. Our results do support that there is correlated motion in the solvation shells around protons upon transfer, in agreement with previous reports. However, as reported in the SI and above in our reply to point 2, there is no effect of proton-proton or proton-chloride correlations on our difference spectra. Most importantly, the difference spectrum for 6M HCl in Fig. 3c looks very similar to the spectrum of local H_5O_2^+ clusters and to the spectrum of the excess proton itself. This means that although there is spatial correlation between excess protons and water clusters, that correlation does not have a spectroscopic signature, which is an important finding.

4. The distribution of ROO of crossing in Figure 4D may not be fair, because the most probable distribution of ROO must be peaked at $\text{ROO} > 2.42 \text{ \AA}$, regardless of PT. Thus, it may be of more interest to compare the probability of crossing at ROO weighted by the total probability at ROO. Also, non-barrier crossing event is often observed in molecular dynamics, from which the 2D (d, ROO) free energy map is statistically averaged. I do not see the reason to use this analysis, or the 2D (d, RROO) free energy map, to support stochastic model, which is kinetic in nature. By the way, I find the successive shoulders in Figure 4D may be of interest because they may favor PT event at these specific O—O separations.

We thank the reviewer for this comment and attempt to clarify. In Figure 1 in this reply we show the distribution of oxygen-oxygen separations R_{OO} of local transient H_5O_2^+ clusters at which the excess proton crosses the mid plane between the water oxygens at $d = 0$ during a complete transfer path (red solid line), this is the same data shown in fig. 4d in our manuscript. We compare with the equilibrium distribution of oxygen-oxygen separations R_{OO} for $d = 0$ (blue line), which is peaked at $R_{\text{OO}} = 2.42 \text{ \AA}$ (right black vertical line) and with the equilibrium distribution for unconstrained d (gray line), which is peaked at $R_{\text{OO}} = 2.51 \text{ \AA}$. Indeed, the complete transfer-path distribution (red line) is shifted to lower values than the unconstrained distribution at $d = 0$ (blue line), which reflects that complete transfer paths are different from other proton paths that cross the midplane.

As correctly pointed out by the referee, many transfer paths are actually barrier-less crossings of the midplane. However, most complete transfer paths, precisely 77%, cross the midplane for $R_{\text{OO}} > 2.39 \text{ \AA}$ (left black vertical line) and therefore experience a finite barrier along d . From the presence of a barrier it follows that the dynamics of such transfer paths and their spectral

signature cannot be captured by a normal mode analysis. We added a discussion of the fraction of transfer paths that experience a finite barrier along d in the paper and also clearly point out that not all transfer paths involve the crossing of a barrier.

We note that the 2D free energy landscape allows us to decompose continuous excess-proton trajectories into segments where the midplane between the flanking water molecules at $d=0$ is crossed and segments that rather correspond to vibrations within local transient H_5O_2^+ clusters. In the revised version of our manuscript we chose different trajectories in Fig. 4e/g and Fig.5a-d in order to clearly bring out how the excess-proton trajectories are decomposed based on the 2D free-energy landscape.

Figure 1 Distribution of R_{OO} positions at which complete transfer paths cross the midplane between the water oxygens at $d = 0$ (red solid line, copy of fig. 4d of the manuscript). For comparison the equilibrium distribution at $d = 0$ is shown in blue and the distribution for arbitrary d in gray.

5. It is of interest to note from Figure 3c that all the features of the difference spectra in the bulk also appear in the difference spectrum between Zundel and two waters. To the reviewer, such observation seems to indicate that none of the bands in the infrared different spectra corresponds to the PT event, because such event does not actually exist in Zundel. On the other hand, the characteristic band of such event may be buried in the signal of proton motions, either bonded or excess.

We thank the reviewer for this comment, which prompted us to clarify the nomenclature used in our manuscript. In literature, the term 'Zundel' usually refers to H_5O_2^+ cations with small oxygen-oxygen separation. However, the local H_5O_2^+ clusters extracted from our HCl simulations exhibit also large oxygen-oxygen separations and pronounced excess-proton asymmetries, as indicated in the 2D free-energy landscape in Fig. 4a. In the initial version of our manuscript, we used the term 'symmetric Zundel state' to refer to a situation where the excess proton is symmetrically shared between two water molecules. Note that Yu et al. [2] very recently used the term 'extreme Zundel' to refer to such a structure. In the new version of our manuscript, we avoid the term 'Zundel' when we talk about local transient H_5O_2^+ clusters. The difference spectrum of H_5O_2^+ clusters and hydrogen-bonded pairs of water molecules, gray line in fig. 3c in the manuscript, includes the complete dynamics of the excess proton and its nearest two water molecules. Within this cluster, proton transfer events between the water molecules happen,

which follows from the simulation trajectory in Fig. 5a-d and which is the reason why the difference spectra of HCl bulk solutions and H_5O_2^+ clusters in Fig. 3c are so similar. As a side remark, in a recent preprint we analyze proton transfer dynamics in isolated H_5O_2^+ cations [1].

[1] Brünig, F. N., Hillmann, P., Kim, W. K., Daldrop, J. O. & Netz, R. R. Proton-transfer spectroscopy beyond the normal-mode scenario. (2021), arXiv:2109.08514.

[2] Yu, Q., Carpenter, W. B., Lewis, N. H. C., Tokmakoff, A. & Bowman, J. M. High-Level VSCF/VCI Calculations Decode the Vibrational Spectrum of the Aqueous Proton. *J. Phys. Chem. B* 123, 7214–7224 (2019).

Reviewer #2 (Remarks to the Author):

I have read the article entitled, "Spectral signatures of stochastic proton transfer events in aqueous hydrochloric acid solutions" with great interest. This is a beautiful piece of theoretical/experimental work. This is my issue with this manuscript. This sentence, "While the stochastic time scales of aqueous proton-transfer events have been studied recently [36], their identification in experimental spectra is a main result of this work." There is then no comparison with the work of ref [36] in accounting for the time scales and mechanisms. Ref [36] appears to be doing something similar--but this paper clearly goes beyond this reference. The question to this reviewer is if this work supersedes the work of Ref [36] or builds on it. This is important to clarify. This manuscript is certainly well done and has incredible analysis. But what is missing is a comparison to other high-quality science (cited, but not discussed). This topic is important and I believe with such a high profile article, the onus is on these authors to put their work in the proper context. Perhaps a statement on what is in agreement with [36] and what is in disagreement with [36].

We thank the reviewer for these very appropriate questions. Ref. [36] in the previous version of our manuscript investigates proton-transfer dynamics in *ab initio* simulations of aqueous HCl solutions and compares with rate theory. Our paper complements that work in that we decompose the absorption spectrum into contributions that have to do with transfer paths and with segments where the excess protons wait in a metastable state. We have added the new section IX to the SI, where we compare our proton transfer waiting times to different time scales from literature (including Ref. 36 in the previous version of our manuscript). A discussion of this comparison was added to our manuscript in the Conclusions.

Please comment on the work of <https://pubs.acs.org/doi/10.1021/jp501091h> (not cited) needs to be brought to light. The finding of a correlated structure in acids is important. To the extent that the counter-ion is not a spectator should be captured (somehow) in this study. This goes well beyond HCl and will be important for other acids (such as the oxyanions) that also show this correlated behavior.

We thank the reviewer for this comment, which prompted us to extract spatial correlation functions from our simulations. We favorably compare various radial distribution functions to results from the previous study [1], suggested by the reviewer, in the new section IV in the SI and briefly mention this comparison in the Introduction as well as in a newly added subsection just before the Conclusions.

[1] Baer, M. D., Fulton, J. L., Balasubramanian, M., Schenter, G. K. & Mundy, C. J. Persistent ion pairing in aqueous hydrochloric acid. *J. Phys. Chem. B* 118, 7211–7220 (2014).

I urge the authors to put this excellent study work in proper context. This is important because the same code was used (CP2K) and a similar reactive picture was used. This work is clearly novel and goes beyond the original experimental work of Tokmakoff (ref 14) and corroborates and provides a reinterpretation of many of the details in his pioneering study. The new experiments are exciting. Having some consistency between the theoretical concepts/simulations would be useful for the community.

If the authors can put their work in proper context of published work in concentrated acids, then this work will be an excellent contribution to the Nature Communications.

As a reaction to this comment and similar comments by the other reviewers, we have added in total four new sections to the SI, where we address diffusion properties of the excess proton (section X), structural correlations around the excess proton (section IV), the role of the hydrogen-bond network around the excess proton and the role of a fourth water molecule that hydrogen-bonds to hydronium (section XI) and where we discuss in detail the proton-transfer waiting-time distributions (section IX). These new sections contain in-depth comparisons to previously published results for concentrated aqueous HCl solutions and allow us to validate our simulations as well as our methodology. We have added the main conclusions from these analyses in the revised version of our manuscript and thereby hopefully have put our paper properly into the context of the existing literature.

Reviewer #3 (Remarks to the Author):

The manuscript "Spectral signatures of stochastic proton transfer events in aqueous hydrochloric acid" by Brünig et al. presents the results of a joint experimental and computational investigation of aqueous solutions of hydrochloric acid and their vibrational spectra. In my opinion, this manuscript is unfortunately not suitable for publication in Nature Communications. After some modifications, it could be suitable for publication elsewhere. Below, I explain my reasons for this conclusion.

A large part of the work focuses on the molecular dynamics simulations and on the analysis of the resulting trajectories - the identification of the excess protons and assignment of spectral features. These molecular dynamics simulations are performed with a classical, rather than quantum, description of the nuclei, and one particular GGA functional with a dispersion correction. The results are then extensively analyzed and the identified spectral features are matched to observed experimental features.

My major concern is that for an acid solution, classical nuclei are a major approximation that is not appropriate for many properties, spectroscopic or others, of these systems. In simulations that do include nuclear quantum effects, results would be substantially different from those presented in this paper. In that case, much of the very detailed analysis would have to be performed again, or in fact re-designed so that the quantum delocalization of the nuclei could be included in a meaningful way. It is not clear at this point which, if any, of the results and their interpretation in the presented manuscript would remain valid if this was done. As the manuscript stands, nuclear quantum effects are mentioned once in the Results section:

"While simulations including quantum-mechanical treatment of the nuclei may be the more accurate model [38, 39], the computationally cheaper Born-Oppenheimer approach was taken in favor of improved statistics for the stochastic analysis."

This seems to constitute a considerable logical error - it is in principle possible that even if a description of nuclear quantum effects would be prohibitively expensive, it would still be crucial to say anything reliable about the system of interest. It is up to the authors to convince the reader that they can draw useful conclusions from their classical simulations for this system for which the importance of nuclear quantum effects is well known. The authors have not even attempted to convince the reader that this is the case. A manuscript that does make such an attempt would have to be considered again, but it seems this would mean major changes throughout the work. I also note that the distinction between a "Born-Oppenheimer approach" and "quantum-mechanical treatment of the nuclei" in the above quoted sentence is wrong. Nuclei can be treated quantum mechanically under the Born-Oppenheimer approximation, and, in fact, that is exactly what imaginary time path integral simulations do - quite a common approach to including NQEs in aqueous solutions. This makes me worry if the authors appreciate what kind of quantum effects come into play in their system and how.

NQEs are mentioned again in Conclusions, noting that including them stabilizes Zundel configurations compared to Eigen. I find it very strange that the authors would say that their

main result is not dramatically modified by this. Their analysis is centered around the characteristic V-shaped effective potential for the excess proton, and if this were to turn into a single-well potential with quantum delocalization, it is not clear, what aspects, if any, of the analysis would be applicable and if the conclusions would be the same. This would certainly have to be shown explicitly, rather than asserted at the very end of the manuscript. The authors even cite a published paper that does include NQEs and uses projection to interpret vibrational spectra, but they do not try to compare to this work.

As such, I find the way the authors approach their neglect of nuclear quantum effects unconvincing and consider it a major issue with the current manuscript.

We thank the reviewer for the remarks concerning the lack of nuclear quantum effects (NQEs) in our simulations. NQEs become relevant when the zero-point energy at frequency ω exceeds thermal energy, $\hbar\omega/2 > k_B T$ [1]. For the lightest nucleus, the proton, in the infrared regime this becomes relevant at room temperature, as has been demonstrated in numerous works. Only recently, NQE simulation techniques have advanced significantly and can be accounted for in molecular dynamics simulations. While it would be interesting to apply our presented trajectory-decomposition techniques to simulations including NQEs, the benefit is unclear at present due to the following reasons:

1) Based on the Trotter formalism, NQEs are typically addressed by replacing each atomic nucleus by a closed-loop polymer, or ring polymer, consisting of P elementary beads, and then running replica simulations. However, the required number of P increases with the maximal frequency of the system. We quote “ $P = 32$ replicas are needed to converge simple structural properties of a system at room temperature containing O–H covalent bonds” [1]. This is the naive factor by which the computational cost would increase (though one should add that replicas can be run in parallel). However, our simulation technique, ab initio molecular-dynamics using DFT, is computationally already very expensive. The total computational time of the HCl simulations performed for our study amounts to 63 days on a high-performance architecture using 394 cores. Additionally, 112 days on 32 cores were used for analysis of the simulation trajectories. The pure-water simulations needed 96 days on 144 cores. Thus, as the reviewer correctly points out, the inclusion of NQEs may be “prohibitively expensive”.

2) Only few studies have so far used NQEs for comparable systems, which was only possible by using additional approximations, for example by reducing the number of replicas P , the so-called ring-polymer contraction (RPC). However, it transpires from the literature that there is not yet a well-established and generally applicable technique, rather, each specific system requires careful optimization of approximations to allow simulations including NQEs with affordable computational cost. This is illustrated in Figure 2, where we summarize various acceleration techniques that have been used to simulate NQEs.

Features	QT	PI+GLE	High-order	RPC and MTS
Applicable to any potential	+	+	– (Explicit), + (RW, FD, PPI) (a)	* (Needs reference potential) (b)
Efficient sampling	– (Overdamped) (c)	+	– (RW), + (FD, PPI) (d)	+
Dynamical properties	* (Reconstruction) (e)	–	–	* (Approximate) (f)
Well-defined ensemble (g)	–	–	+	+
Suitable for all estimators	–	* (Custom fits) (h)	+	+
Physically meaningful momenta	* (Approximate) (i)	–	–	–
Typical error in NQEs, % (j)	5–10	1–2	<1	1–2

+, a positive of the technique; –, a deficiency of the technique.

Figure 2: Summary of different acceleration methods for NQEs, taken from [1].

3) Simulations of dynamical properties including NQEs remain a “challenging problem” [1], as becomes evident from Figure 2. Strictly speaking, treating NQEs on the basis of the Trotter formalism is only valid for static observables. So far, to obtain dynamical properties, NQE simulations have been mostly propagated using either centroid molecular dynamics (CMD) or ring-polymer molecular dynamics (RPMD). Note, that both methods are based on ring polymers and do not include correlations between nuclei and electrons. We therefore agree with the reviewer that current simulations with and without NQEs both rely on the Born-Oppenheimer approximation. Still, “the formal justification of both CMD and RPMD does not involve a hierarchy of well-controlled approximations starting from the full quantum mechanical expression for the various time correlation functions. Because one cannot identify or compute terms that are neglected by these methods, it is hard to systematically address their known artefacts, which becomes particularly problematic at low temperatures or when calculating the nonlinear operators encountered in many types of spectroscopy” [1]. Indeed, different NQE methods typically lead to different results, as exemplified in Figure 3, copied from reference [2], which shows IR spectra for the Zundel cation, H_5O_2^+ , in vacuum obtained from different NQE simulation techniques. We mention that diffusion coefficients are also known to be vastly affected by different methods for implementing NQEs [3].

Figure 3: IR spectra for the Zundel cation in gas phase, $H_5O_2^+$, from various NQE simulation techniques: RPMD, CMD and thermostatted RPMD (TRPMD). 'Classical' refers to data that is obtained without considering NQEs. Figure taken from [2].

4) There is, to our knowledge, a single study of aqueous HCl solutions that shows IR spectra obtained from ab initio molecular dynamics simulations on a comparable level of DFT accuracy as employed in our simulations, but in addition includes NQEs [4]. This is achieved by making relatively drastic simplifications, for example by performing ring-polymer contraction to a single bead. As a side remark, the authors perform most of their analysis, which focuses on the hydrogen-bond dynamics, based on trajectories that are obtained without NQEs. Nevertheless, this study is important as it compares IR spectra obtained from simulations with and without NQEs for a similar DFT level. The spectra from that study, which also includes a comparison to experimental data, are reproduced in Figure 4. The differences between the absolute spectra and between the difference spectra obtained with and without NQEs are actually negligible, except for high frequencies of the OH stretch vibration around 3300 cm^{-1} . This high-frequency regime is not reporting on excess-proton dynamics and therefore not relevant for our study. In contrast, the study finds large deviations for the mean distribution of the excess-proton positions along a proton sharing coordinate between simulations with and without NQEs, due to zero-point motion. While the strong effect of NQEs on spatial excess-proton distributions has been known for a long time [5], the effect of NQEs on excess-proton dynamics seems to be rather small. This conclusion is corroborated by IR difference spectra reported in another study, comparing spectra with and without NQEs for two versions of the multistate empirical valence bond (MS-EVB 3.2) model for the hydrated excess proton [6]. The spectra taken from [6] are shown in Figure 5 and likewise do not differ much between simulations with and without NQEs. Another recent study performed simulations of a single HCl pair in water using a comparable DFT model and accounted for NQEs [7]. The study addressed static properties using a very expensive path-integral simulation with 30 beads for 32 ps and calculated dynamical properties by RPC simulations with a reference potential that was obtained using machine-learning

techniques. Also from this study, it transpires that either extensive computational resources or advanced methods and uncontrolled approximations are required to account for NQEs in a computationally feasible fashion.

Figure 4: IR and VDOS spectra and difference spectra for 4M HCl solution from experiment and *ab initio* molecular dynamics simulations with quantum and classical nuclei, taken from [4].

Figure 5: IR difference spectra obtained from classical and CMD trajectories of (a, left) aMS-EVB 3.2 and (b, right) MS-EVB 3.2 simulations of 1 HCl aqueous system along with the experimental attenuated total reflection (ATR) difference spectrum [6].

5) Generally, NQEs become less important at higher temperatures and lower wavenumbers. Our work focuses on the proton dynamics in the THz regime and at room temperature, in fact, we compare our simulations to experimental results in a frequency range down to 100 cm^{-1} .

Presumably, NQEs should be even less relevant in this frequency range.

In conclusion, our chosen simulation technique, ab initio DFT simulations with the BLYP exchange-correlation functional, is well established and widely used for studying excess proton dynamics in water. From our arguments above we conclude that NQEs are less important for dynamical excess-proton properties and spectra than they are for static spatial distribution properties. Therefore, we expect our results, that focus on dynamical excess-proton properties, to be robust with respect to nuclear quantum effects. We have added an according statement and a detailed discussion along these lines in a newly added section just before the Discussion section.

[1] Markland, T. E. & Ceriotti, M. Nuclear quantum effects enter the mainstream. *Nat. Rev. Chem.* 2, (2018).

[2] Rossi, M., Ceriotti, M. & Manolopoulos, D. E. How to remove the spurious resonances from ring polymer molecular dynamics. *J. Chem. Phys.* 140, 234116 (2014).

[3] Marsalek, O. & Markland, T. E. Quantum dynamics and spectroscopy of ab Initio liquid water: The interplay of nuclear and electronic quantum effects. *J. Phys. Chem. Lett.* 8, 1545–1551 (2017).

[4] Napoli, J. A., Marsalek, O. & Markland, T. E. Decoding the spectroscopic features and time scales of aqueous proton defects. *J. Chem. Phys.* 148, 222833 (2018).

[5] Marx, D., Tuckerman, M. E., Hutter, J. & Parrinello, M. The nature of the hydrated excess proton in water. *Nature* 397, 601–604 (1999).

[6] Biswas, R., Tse, Y. L. S., Tokmakoff, A. & Voth, G. A. Role of Presolvation and Anharmonicity in Aqueous Phase Hydrated Proton Solvation and Transport. *J. Phys. Chem. B* 120, 1793–1804 (2016).

[7] Calio, P. B., Li, C. & Voth, G. A. Resolving the Structural Debate for the Hydrated Excess Proton in Water. *J. Am. Chem. Soc.* 143, 18672–18683 (2021).

Even with classical nuclei, some of the analysis relies on specific features of the shape of the effective potential. If this were to change with a different density functional, would the analysis remain the same? This is unclear from the text. It is known in the literature that the combination of classical nuclei and GGA functionals results in a cancellation of errors that yields surprisingly good results for some quantities, compared to experiment. Regardless of how this cancellation still does or does not apply to systems with proton defects, a good match to experiment at this level of methodology should be viewed with a healthy degree of skepticism. The authors take the results they get with their particular functional and classical nuclei at face value note a good match with experiment.

We agree with the referee and added a statement in the Introduction, saying that the good agreement between spectra from our simulations and experiments could be due to cancellation of errors.

As a side remark to the referee, even if cancellation of errors is at work, this would mean that the temporal two-point correlation functions of the polarization from simulations are accurate, from which one could conclude that an in-depth analysis of these correlation functions is meaningful.

The projection analysis of vibrational spectra itself is potentially interesting and helpful for interpreting experiments, if it could be presented in a way that is general for any proton defect, rather than relying on the existence of specific minima or barriers in the effective potential for the excess proton. One thing I am missing in this regard is an explanation of how this effective potential varies with the changing broader surrounding of the defect in the solution, again something that has been discussed in the literature before. This is important when talking about specific trajectories, such as the ones shown as the various examples, because these most likely do not evolve in the average potential, but rather an instantaneous distortion of it. If these distortions evolve on a time scale similar to or slower than the proton transfer trajectories analyzed here (and from the literature that appears to be the case), talking about the trajectory crossing specific barriers or other landmarks in the average potential might be tricky.

To address similar questions by referees 1 and 2, we have analyzed the coupling of the excess-proton dynamics to its solvation environment and have described the results of these analyses in the newly added sections IV and XI in the SI, together with a short discussion in the Introduction and in a newly added subsection just before the Conclusion section in the manuscript. Our decomposition of the excess-proton trajectories is indeed adapted to the 2D distribution of the excess proton in terms of the oxygen-oxygen separation and the proton asymmetry in a local transient H_5O_2^+ complex, for other systems a different decomposition might be more appropriate. The main point of our work is that the decomposition of an excess-proton trajectory based on free-energy features allows to decompose the absorption spectrum and thereby to interpret the spectrum contributions of transfer events that cannot be treated by normal-mode analysis.

One could also define time-dependent probability distributions based on features in the time-dependent trajectories. Our strategy is reverse, we decompose the excess-proton trajectory into segments based on features in the time-independent 2D distribution function, an approach that is borrowed from the non-equilibrium statistical mechanics of barrier-crossing in multi-dimensional systems. We mention that our methodology does not depend on the existence of barriers in the free-energy landscape and can be done based on arbitrary separatrix lines or separatrix surfaces in a multidimensional state space. The value of our decomposition scheme is appreciated a posteriori, since the decomposed trajectories project out distinct peaks in the absorption spectrum. We changed the title of our paper and added explanations to our paper that describe what we believe is the main advantage of our methodology.

The distinction between Eigen and Zundel configurations based on the d coordinate (called δ or ν in the existing literature, by the way) and saying that Zundel has $d=0$ seems very simplistic and not useful.

We agree with the reviewer, that our wording in the previous version of our paper was potentially confusing. One could call the configuration with $d=0$ a 'symmetric Zundel state'. Similarly, Yu et al. [1] very recently used the term 'extreme Zundel' to refer to that state. Instead of such terminology, in the revised version of the manuscript, we avoid the terms 'Zundel' and 'Eigen' when discussing our data and only use these terms when referring to literature.

[1] Yu, Q., Carpenter, W. B., Lewis, N. H. C., Tokmakoff, A. & Bowman, J. M. High-Level VSCF/VCI Calculations Decode the Vibrational Spectrum of the Aqueous Proton. *J. Phys. Chem. B* 123, 7214–7224 (2019).

The electronic structure setup seems reasonable, though some benchmarking of the short-range double-zeta molopt basis might be warranted, perhaps compared to the non-short-range double-zeta molopt and non-molopt triple-zeta basis sets.

As suggested by the reviewer, we performed additional simulations of HCl solutions at 6M with the non-short-range molopt basis set and the non-molopt TZV2P basis set and compare the spatial correlations in terms of radial distribution functions in the additional subsection IV.3 in the SI. The new data shows a slight increase of the coordination of excess protons with chloride ions. However, the coordination of excess protons with oxygen nuclei, which is the dominant solvation mode and the focus of our study, does not change appreciably when using a different basis set.

There are other, smaller and more specific, comments that could be made on this paper, but that seems premature until the above major concerns are addressed.

We hope that we could address the referee's concerns sufficiently and welcome further suggestions.

Reviewer #4 (Remarks to the Author):

In this manuscript, the authors use ab initio molecular dynamics to compute the IR spectrum of the aqueous proton from a dynamical perspective instead of typical normal mode analyses. Despite decades of experimental and theoretical efforts, the physical, dynamical, and spectroscopic properties of the aqueous proton remain incredibly challenging to pin down. There has been a recent resurgence in this area thanks to rapid advancements in experimental (particularly ultrafast IR) and computational methods, but many questions remain on the interpretation of the IR and 2D IR spectra of the aqueous proton. The dynamical perspective presented in this manuscript is a novel contribution to the many studies on the aqueous proton.

The manuscript as presented, in my opinion, requires some major revisions before I can fully endorse publication. My comments/suggestions/concerns are presented below:

1. I think there is an excellent opportunity to discuss and present the dynamical perspective given here with both the normal mode and experimental interpretations recently published by Tokmakoff and Bowman (JPCB, 2019, 123, 7214 and JCP, 2020, 153, 124506) which I believe is the current standard for our understanding of the aqueous proton. Somehow these papers are not even cited.

I do not support the dismissive attitude towards normal mode approaches, especially the sentence in the Introduction, “normal-mode approximations, which however fail, for an obvious reasons”. I agree that there are issues with the normal mode perspective, but I see the normal mode perspective and this new dynamical perspective as highly complementary and I think they should be presented as such. In fact, the normal mode perspectives of Tokmakoff and Bowman have been quite successful in interpreting many of the features in the IR and, more impressively, 2D IR spectra. For example, the “superharmonic” behavior of the shared proton stretch at 1200 cm^{-1} and the bending motion of the aqueous proton complex near 1750 cm^{-1} , which is highly coupled and mixed with the proton stretch.

We agree with the reviewer that normal-mode calculations have been and will remain at the basis of the interpretation of vibrational spectra of molecules and liquids. Our wording indeed was misleading and we revised our manuscript accordingly. Our main point is that IR-active barrier-crossing events, so-called ‘unstable’ modes, cannot be described by normal modes but lead to characteristic spectral signatures that have not been treated in the literature discussion so far. In fact, in the revised version of our manuscript we now explicitly say that 77% of all proton-transfer events between two water molecules, namely those that occur for relatively large separation between the water molecules, in fact do involve a barrier crossing. We also demonstrate that the stochastic theory of activated rate processes is well suited for the theoretical treatment of the spectral signatures of such processes.

In line with the referee’s comments, we find in our analysis that one of the main IR spectral signatures of the excess proton, the continuum band between the bending and stretching vibrations of liquid water, corresponds to normal modes in a rapidly changing potential. We mentioned this already in the initial version of the manuscript but did not refer to the works

suggested by the reviewer. Indeed, the recent work by Tokmakoff and Bowman [1,2] presents itself highly complementary to our results. We elaborate on this in detail in the following.

[1] Yu, Q., Carpenter, W. B., Lewis, N. H. C., Tokmakoff, A. & Bowman, J. M. High-Level VSCF/VCI Calculations Decode the Vibrational Spectrum of the Aqueous Proton. *J. Phys. Chem. B* 123, 7214–7224 (2019).

[2] Carpenter, W. B. et al. Decoding the 2D IR spectrum of the aqueous proton with high-level VSCF/VCI calculations. *J. Chem. Phys.* 153, 124506 (2020).

2. Along this line, the current paper doesn't address the acid bending mode at 1750 cm^{-1} . A feature in this region isn't clearly reproduced by the simulations, perhaps at higher concentrations. Since the proton stretch and bend are highly coupled and mixed, I would anticipate the "transition path" dynamics that give rise to the 1200 cm^{-1} feature should also contribute to the acid bending region.

We thank the reviewer for this comment, which led us to discuss the acid-bend signature in the revised version of our manuscript. The acid bending mode around 1750 cm^{-1} is an important spectral signature that is observed throughout experimental and simulation data of excess protons in water. It appears prominently in IR difference spectra due to a blue shift of the bending modes of the protonated water in comparison to the 1650 cm^{-1} bending mode of unprotonated water (see summary of experimental IR spectra in fig. S1 in the SI) and has been the focus of numerous experimental studies. In our simulated difference spectra, the acid band is not clearly distinguishable from the continuum band, in agreement with previous simulations (see fig. 4b in the reply to reviewer 3 above).

However, a careful analysis of the vibrational spectrum of the excess proton along the oxygen-oxygen axis in fig. 5e of our manuscript (a copy of the figure with a blow up of the relevant regime is provided below in Figure 6) reveals distinct but small peaks in the spectral decomposition into transfer-waiting (TW), transfer-path (TP) and normal-mode (NM) contributions in the range between 1400 cm^{-1} and 1800 cm^{-1} . The TW contribution peaks at about 1650 cm^{-1} , consistent with a peak in the spectrum of the R_{∞} coordinate in the lower panel of fig. 5e in our manuscript, indicating that this signature is related to motion of the neighboring oxygen atoms and therefore hinting to a weak coupling to an unperturbed water bending mode. The TP contribution peaks indeed at 1750 cm^{-1} , the location of the acid bend, which confirms the suggestion by the reviewer and indicates that the acid bend couples particularly to the transfer path of the excess-proton transfer. The NM contribution peaks at about 1500 cm^{-1} , which is a feature that is also visible in the Eigen-like spectra derived by Yu et al. [1].

We have added a new paragraph on page 10 of the manuscript where we discuss the acid bend signature with respect to our decomposition and clearly distinguish the acid band from the continuum band throughout the revised version of our manuscript.

[1] Yu, Q., Carpenter, W. B., Lewis, N. H. C., Tokmakoff, A. & Bowman, J. M. High-Level VSCF/VCI Calculations Decode the Vibrational Spectrum of the Aqueous Proton. *J. Phys. Chem. B* 123, 7214–7224 (2019).

[2] Napoli, J. A., Marsalek, O. & Markland, T. E. Decoding the spectroscopic features and time scales of aqueous proton defects. *J. Chem. Phys.* 148, 222833 (2018).

Figure 6: Copy of fig. 5e from the manuscript, with a blow up of the regime around 1000 to 3000 cm^{-1} .

3. I have some questions/concerns regarding the definition of “proton transfer events”. The authors state “The actual proton transfer involves the thermally activated crossing of an energy barrier...” I agree with this statement, but the timescales of the ‘barrier crossing’ event do not match up with experiments. Polarization anisotropy measurements by Tokmakoff (JPCB, 2018, 122, 2792 and JCP, 2019, 151, 034501) showed a slow 2.5 ps decay and was interpreted in terms of irreversible proton transfer (not just rattling) that requires overall reorganization of the local H-bond network, in agreement with NMR measurements and known H-bond switching timescales in water. This is the barrier that must be crossed for proton transfer.

In my opinion, the fast dynamics captured in these simulations are not irreversible Grotthuss proton transfer events but rather fast rattling of the proton within the proton complex (either between a special pair in a Zundel picture or Greg Voth special pair dance in an Eigen picture). The proton stretch 2D IR spectra of Elsaesser and Tokmakoff are consistent with a low-barrier proton stretch potential where the zero-point level lies above the barrier between proton donor and acceptor. Therefore, the proton wavefunction is highly delocalized and hence the interpretation in terms of more Zundel-like configurations.

2D IR supports a relatively localized and persistent Zundel-like complex despite fast H-bond fluctuations. How do these recent experimental results/interpretations fit into the presented simulations? On what timescales are these simulations capturing irreversible transfer and H bond rearrangements? Are these simulations really only tracking fast rattling/special pair dancing? If so, I would not use the phrase “proton transfer” but rather “proton dynamics”.

We thank the reviewer for this comment, which prompted us to compare our proton-transfer waiting time distributions with other descriptors of proton-transfer dynamics. We find that our proton-transfer waiting-time distributions match very well the long time scale of the hydronium continuous-identity auto-correlation function, introduced in [1], which thus is shown to be an equivalent observable. If fast recrossings, i.e., back-and-forth proton transfer events or proton rattling, are removed from the trajectories prior to the analysis, the hydronium continuous-identity auto-correlation function decays on time scales that are comparable to the 1-2 ps reported in the literature for irreversible proton transfer and related to the time scales of reorganization of the hydrogen bonds in water. We discuss these results in the new section IX of the SI.

However, these long time scales will only lead to spectroscopic features below 1 THz, while the spectroscopic features above 1THz are in fact caused by the fast back-and-forth proton transfer events.

[1] Arntsen, C., Chen, C., Calio, P. B., Li, C. & Voth, G. A. The hopping mechanism of the hydrated excess proton and its contribution to proton diffusion in water. *J. Chem. Phys.* 154, 194506 (2021).

4. The authors admit that their simulations support a more Eigen-centric picture, in contrast to the experimental interpretations discussed above, but state that an Eigen or Zundel picture wouldn't change the results. This makes me concerned that the simulations are getting the right answers (or at least reasonable agreement with experiment) for the wrong reasons.

Tokmakoff and Bowman have been careful in their definition of Eigen and Zundel. Although there is a large distribution of O-O and O-H distances that span traditional Eigen and Zundel definitions, the 2D IR spectra support a more localized special pair arrangement, i.e., and strongly bound proton between two unique, special water molecules. Hence, the description of the complex as "Zundel-like". Using these more relaxed definitions, do the simulations presented here support this view, or are they really more Eigen-centric? The presented PES's indeed seem more Eigen centric and the large barrier to proton shuttling between nearest neighbors is a bit concerning given the current view of the aqueous proton.

We agree that a distinction of the solvated proton into Eigen and Zundel configuration is not helpful and therefore avoid these terms when referring to our data in the new version of our manuscript. It is generally known that the type of DFT simulations we perform tends to favor Eigen-like state, while the inclusion of nuclear-quantum effects (NQEs) favors Zundel-like states. However, spectroscopic features in the mid-IR regime are not changed much upon the inclusion of NQEs [1], suggesting that NQEs change excess proton dynamics less than they change excess-proton equilibrium distribution (see also our replies to reviewer 3 above). The main point is that if a Zundel-like state play the role of a transition state, it will occur with a small probability but will nevertheless make a sizeable spectroscopic contribution, since the proton moves quickly and over large distances when it crosses a barrier. Such a contribution is missed by a normal-mode analysis, which explains why our transfer-path analysis is important for correctly interpreting experimental and simulated spectra. We added comments along these lines on pages 2 and 11 of the manuscript.

[1] Napoli, J. A., Marsalek, O. & Markland, T. E. Decoding the spectroscopic features and time scales of aqueous proton defects. *J. Chem. Phys.* 148, 222833 (2018).

5. One place where the normal mode picture does fail is in the continuum region near 2500 cm^{-1} . 2D IR spectra show all the spectral signatures of the aqueous proton being highly coupled, pointing to a single and indistinguishable Zundel-like proton complex. Normal mode analyses predict this region to arise from more Eigen-like OH stretches and predict weak coupling to the proton stretch and acid bend modes. Tokmakoff and Bowman suggest this difference could be from fast fluctuations of the complex that can't be captured in the normal mode approach. Can the authors provide any more insight into proton stretch-bend-continuum coupling from the dynamical perspective that could help interpret the experimental 2D IR observations?

Our dynamic decomposition shows that the continuum band between 2000 cm^{-1} and 3000 cm^{-1} is described by the “normal-mode” (NM) contribution and therefore corresponds to stable motion of the excess-proton around local minima along the proton sharing coordinate d of our 2D free energy landscape. Importantly, the NM contributions are not part of transfer paths (TP) and many regions of the IR spectrum can from our perspective be rightfully modeled by a local normal-mode approximation with appropriate anharmonic corrections, as was carefully done by Tokmakoff and Bowman. However, in our decomposition, the NM contributions (whose spectral signature corresponds to Eigen-like complexes in the normal-mode picture by Tokmakoff and Bowman [1]) are alternating with the TP contributions (which produce the 1200 cm^{-1} signature associated with Zundel-like complexes in the normal-mode picture by Tokmakoff and Bowman and also others) on a stochastic time scale. That stochastic time scale is the transfer-waiting time. Since the transfer-waiting time exhibits an exponential distribution, as shown in Fig. 5f in our manuscript, fast alternation between local Eigen-like complexes and Zundel-like complexes is indeed supported by our analysis and perfectly confirms the suggestion by Tokmakoff and Bowman quoted by the reviewer. This very broadly distributed interconversion time may also be the reason for the strong coupling observed in 2D IR experiments between the spectral signatures of the excess proton and in particular between distinct Zundel-like and Eigen-like complexes in the normal-mode picture. Once again, this shows that analyzing spectral signatures in terms of normal modes is difficult when barrier crossing is involved.

[1] Yu, Q., Carpenter, W. B., Lewis, N. H. C., Tokmakoff, A. & Bowman, J. M. High-Level VSCF/VCI Calculations Decode the Vibrational Spectrum of the Aqueous Proton. *J. Phys. Chem. B* 123, 7214–7224 (2019).

In summary, I would suggest that the authors discuss their results in the context of the normal mode and experimental analyses of Bowman, Tokmakoff, and others, and present their results as complementary and as providing a new perspective. What new insights do these simulations give us? Do they change how we should think about the aqueous proton? Do they offer new or alternative interpretations of experimental data compared to normal mode approaches? Are any new interpretations provided that can't be answered by normal mode approaches?

As explained in our replies to the previous comments, we believe that our analysis

complements and confirms but also goes beyond normal-mode approaches. We find that the continuum band can be described by normal modes. However, it is important that these local structures fluctuate rapidly as part of the proton-transfer process, which shows a broad distribution of waiting times and includes high frequency events. Presumably, these fast interconversions couple distinct normal modes, as suggested by 2D IR spectroscopy.

We predict the spectral signature of the transfer process itself to appear at 1200 cm^{-1} , a regime that in the normal-mode picture is associated with Zundel-like structures. In fact, we find that roughly three quarter of all transfer paths happen for large oxygen-oxygen separations, for which a finite free energy barrier is present and thus a normal-mode picture does not hold, for the remaining one quarter a barrier is absent and a normal-mode picture works.

Furthermore, we make predictions for much lower frequencies in the THz regime, where we expect a signature of the stochastic transfer-waiting times. As Tokmakoff and Bowman themselves state in their work, their normal-mode description is not suitable for low frequencies and they comment little on vibrational signatures below 1200 cm^{-1} , in fact, unstable modes and modes around $500\text{ to }700\text{ cm}^{-1}$ are explicitly excluded. This is precisely the regime we assign to the time scales of proton transfer, which together with state-of-the-art experimental THz spectra is at the focus of our work

We have thoroughly revised the manuscript in order to clarify which spectral contributions can be described by normal-modes and in which spectral ranges transfer events that involve barriers leave their mark. Furthermore, we added a paragraph to the Conclusions, where we compare our findings with the recent results by Tokmakoff and Bowman.

Best wishes,

Prof. Dr. Roland Netz

REVIEWERS' COMMENTS

Reviewer #1 (Remarks to the Author):

In this revision, the authors performed additional study on the aqueous HCl solution at higher concentration with ab initio molecular dynamics simulation. The amount of work is quite extensive. Some of my concerns are well addressed, such as diffusion coefficients, and some are partially addressed, such as the correlated motion of the other water molecules beyond the Zundel-like complex. The following are my further comments regarding the revision.

1. A sentence is expected to be added in the abstract to help the readers to understand the scientific significance of this study.
2. The term “excess proton” or “excess-proton” may not be appropriate. Because all the protons are balanced by counter-ions in this study, there is no “excess” proton in the model systems in the strict sense.
3. Figure 1a may be confusing without interpreting the d-axis, though it is defined in the text and pictorially explained in Figure 3a. I suggest to present that schematic picture in Figure 1a instead.
4. The decomposition of the PT trajectory to the segments of TW, TP, and NM in Figure 5a-5d is the merit of the manuscript. It would be nice if the authors can add some discussions to justify the correlation between the power spectra of the d-degree of freedom to the IR spectra.
5. Equation 16 in SI, the construction of the rotation matrix M needs to be illustrated in order to assist the reader to understand the orientation of the YZ plane.
6. In my opinion, the newly added section, “Alternative methods for simulation and characterization of excess-proton dynamics, may not be very relevant to the main content. I suggest to move this section to SI in order to focus on the discussions on the IR and the decomposed power spectra.

Reviewer #4 (Remarks to the Author):

I commend the authors for their careful and extensive revisions of their manuscript and their well-written rebuttal letter. While I still have some reservations regarding the manuscript, I think the work is an important new contribution to the physical chemistry of the aqueous proton and its transfer dynamics and, as such, is now suitable for publication in Nature Comm.

Prof. Dr. Roland Netz
Arnimallee 14
14195 Berlin

Telefon +49 30 838-55737
Fax +49 30 838-53741
E-Mail rnetz@physik.fu-berlin.de
Internet www.physik.fu-berlin.de/
einrichtungen/ag/ag-netz

09.06.2022

Manuscript "**Spectral signatures of excess-proton waiting and transfer-path dynamics in aqueous hydrochloric acid solutions**"

We appreciate the thoughtful referee reports. In the following, we reproduce the report of referee 1 in full and explain the changes made to our paper. Apart from minor changes, all additions to our paper are in the marked-up version shown in red.

Reviewer #1 (Remarks to the Author):

In this revision, the authors performed additional study on the aqueous HCl solution at higher concentration with ab initio molecular dynamics simulation. The amount of work is quite extensive. Some of my concerns are well addressed, such as diffusion coefficients, and some are partially addressed, such as the correlated motion of the other water molecules beyond the Zundel-like complex. The following are my further comments regarding the revision.

1. A sentence is expected to be added in the abstract to help the readers to understand the scientific significance of this study.

We have shortened and rewritten the abstract in order to comply with the formal requirements of Nature Communications. By this we also tried to more clearly bring out the scientific significance of our study.

2. The term "excess proton" or "excess-proton" may not be appropriate. Because all the protons are balanced by counter-ions in this study, there is no "excess" proton in the model systems in the strict sense.

We understand the referee's comment but note that in all experimental studies using acidic solutions the excess proton charge is neutralized by counterions. In order to define the meaning of the term "excess proton" early on in the paper, we have added a short discussion on page 1 in the Introduction.

3. Figure 1a may be confusing without interpreting the d-axis, though it is defined in the text and pictorially explained in Figure 3a. I suggest to present that schematic picture in Figure 1a instead.

We appreciate the comment and have adapted fig. 1 to include the definition of the d variable.

4. The decomposition of the PT trajectory to the segments of TW, TP, and NM in Figure 5a-5d is the merit of the manuscript. It would be nice if the authors can add some discussions to justify the correlation between the power spectra of the d -degree of freedom to the IR spectra.

This comment refers to a central point in our study. As we show in Fig. 3c, the IR difference spectrum calculated from the entire simulation system agrees with the excess-proton spectrum along the d -degree up to a scaling factor. This shows that the difference spectrum of a HCl solution (simulated or experimental) reports on the excess-proton dynamics, and in turn also means that analysis of the excess-proton dynamics sheds light on the spectroscopic signatures of HCl solutions. We have added a comment on page 6 in the revised version where we also explicitly state that this equivalence constitutes a central validation of our approach.

5. Equation 16 in SI, the construction of the rotation matrix M needs to be illustrated in order to assist the reader to understand the orientation of the YZ plane.

In response to the comment we have adapted Supplementary Fig. 10 as part of Supplementary Note 5 to illustrate the coordinate transform.

6. In my opinion, the newly added section, "Alternative methods for simulation and characterization of excess-proton dynamics, may not be very relevant to the main content. I suggest to move this section to SI in order to focus on the discussions on the IR and the decomposed power spectra.

We agree with this comment and have accordingly moved the section into the Supplementary Information as Supplementary Note 1. In the revised version we refer to the Supplementary note at the end of the Results section.

We sincerely hope that the revised version of the manuscript is acceptable for publication in Nature Communications.

Best wishes,

Prof. Dr. Roland Netz